# It's about time: Analysing simplifying assumptions for modelling multi-step pathways in systems biology

**Niklas Korsbo**[1,2], **Henrik Jönsson**[1,2,3]*

**1** The Sainsbury Laboratory, University of Cambridge, Cambridge, United Kingdom, **2** Department of Applied Mathematics and Theoretical Physics, University of Cambridge, Cambridge, United Kingdom, **3** Department of Astronomy and Theoretical Physics, Computational Biology and Biological Physics, Lund University, Lund, Sweden

* henrik.jonsson@slcu.cam.ac.uk

**Data Availability Statement:** All code and instructions on how to reproduce all results are available at the Sainsbury Laboratory Gitlab Repository https://gitlab.com/slcu/teamhj/publications/korsbo_et_al_2020.

## Abstract

Thoughtful use of simplifying assumptions is crucial to make systems biology models tractable while still representative of the underlying biology. A useful simplification can elucidate the core dynamics of a system. A poorly chosen assumption can, however, either render a model too complicated for making conclusions or it can prevent an otherwise accurate model from describing experimentally observed dynamics. Here, we perform a computational investigation of sequential multi-step pathway models that contain fewer pathway steps than the system they are designed to emulate. We demonstrate when such models will fail to reproduce data and how detrimental truncation of a pathway leads to detectable signatures in model dynamics and its optimised parameters. An alternative assumption is suggested for simplifying such pathways. Rather than assuming a truncated number of pathway steps, we propose to use the assumption that the rates of information propagation along the pathway is homogeneous and, instead, letting the length of the pathway be a free parameter. We first focus on linear pathways that are sequential and have first-order kinetics, and we show how this assumption results in a three-parameter model that consistently outperforms its truncated rival and a delay differential equation alternative in recapitulating observed dynamics. We then show how the proposed assumption allows for similarly terse and effective models of non-linear pathways. Our results provide a foundation for well-informed decision making during model simplifications.

## Author summary

Mathematical modelling can be a highly effective way of condensing our understanding of biological processes and highlight the most important aspects of them. Effective models are based on simplifying assumptions that reduce complexity while still retaining the core dynamics of the original problem. Finding such assumptions is, however, not trivial. In this paper, we explore ways in which one can simplify long chains of simple reactions wherein each step is only dependent on its predecessor. After generating synthetic data

**Funding:** This work was supported by the Gatsby Charitable Foundation (grant GAT3395-PR4B; https://www.gatsby.org.uk/). The funders had no role in study design, data collection and analysis, decision to publish, or preparation of the manuscript.

**Competing interests:** The authors have declared that no competing interests exist.

from models that describe such chains in explicit detail we compare how well different simplifications retain the original dynamics. We show that the most common such simplification, which is to ignore parts of the chain, often renders models unable to account for time delays. However, we also show that when such a simplification has had a detrimental effect it leaves a detectable signature in its optimised parameter values. We also propose an alternative assumption which leads to a highly effective model with only three parameters. By comparing the effects of these simplifying assumptions in thousands of different cases and for different conditions we are able to clearly show when and why one is preferred over the other.

## Introduction

Biochemical reaction networks are often complicated and any attempt to describe them using mathematical models relies heavily on simplifying assumptions [1]. Effective models are often built upon simplifying assumptions that avoid over-fitting by using as few free parameters as possible while still capturing the main properties of the biological system [2, 3]. Thoughtful assumptions, as well as robust methods to identify parameter values, and (semi) global analysis of dynamical behaviour within model spaces, are all essential when evaluating models [4–7]. However, assumptions that are beneficial in one setting may be detrimental in another and it is important, although non-trivial, to identify when this happens [8].

Multi-step processes are ubiquitous in biology. Examples are transcription and translation, where an RNA polymerase or a ribosome can perform thousands of sequential reactions before a protein is produced. Yet, in gene-regulatory networks, this is often reduced to a one or two-step reaction of a transcription factor that may lead to an mRNA before it leads to a finished protein (e.g. [1, 9–11]). Another example is kinase cascades, where the product of a kinase triggers the action of downstream kinases [12]. A well known such cascade can be found in the MAP kinases, which are triggered by the MAP kinase kinases, which in turn are triggered by the MAP kinase kinase kinases [13–18]. There are also molecules which must undergo sequential multi-site phosphorylations before they are activated and can pass on any signalling [19, 20]. This is, for example, important for the Drosophila circadian clock protein CLOCK who's inactivity, activity, and degradation are thought to be governed by its sequential states of phosphorylation [21]. Sequential multi-step reactions can also be important for signal perception and transduction pathways. One example is the receptor-like kinase FLAGELLIN SENSING 2 which, upon detecting of a pathogen, triggers a long chain of phosphotransfers, phosphorylations, and subcellular re-localisations that eventually leads to an immune response in *Arabidopsis thaliana* [22, 23]. Another well studied system is the TGF$\beta$ growth factor, which similarly triggers a sequence of phosphorylation steps before affecting the expression of downstream genes [24–26].

Linear pathways represent a large class of multi-step reactions which are both biologically relevant and theoretically approachable. Multi-step pathways can be represented as a chain of state changes where the activation rate of one state is dependent on the activity of the previous state. The mechanisms by which one active state regulates the activation of the next may be complicated but it can be useful to approximate these as being linear. This is partly because it is a minimally complex assumption and partly because many biochemical reactions appear to be linear as long as they operate in a weakly activated manner, far from saturation [12, 27].

Linear pathways are dynamically important and can entirely change the qualitative behaviour of a model. Their main effects are to supply signal amplification/dampening and to

provide delays in the signalling [12, 28]. The amplitude modulation of the signalling is governed by the ratio between the activation and inactivation rates; a pathway step will provide amplification if its activation rate exceeds its inactivation rate. The time-delay, on the other hand, is governed by the inactivation rates and by the length of the pathways [12, 27]. These time-delays can have a significant impact on how a biological system works. A striking example of this is that delays are required for oscillations to be possible [29–31].

The modelling of linear pathways poses a specific set of challenges. Full enumeration of the linear pathway greatly increases model complexity yet add disproportionately little in terms of dynamical range. However, even if the individual steps are of little dynamical significance, the aggregate effect of the full pathway may not be. There is, therefore, a need for a simplifying assumption which reduces the complexity of the linear pathway while still representing its total effect.

A common way of simplifying linear pathways is to ignore most of the reaction steps and assume that a model can recapitulate their effect using only one or a few steps [3]. Such topological model reduction is common and the approach has been rigorously analysed for both simple and complex networks [32–34]. While important, such analyses often presuppose precise knowledge of the system that is being simplified. Nevertheless, while this assumption is often implicit, it is easy to find examples where it has been used to simplify multi-step reactions in real systems—where many details are unavailable—such as protein production (e.g. [7, 35–37]); protein-to-protein signalling networks (e.g. [7, 36–38]); protein modifications such as phosphorylations, methylations, and ubiquitinations (e.g. [37, 38]); and more. A question that remains is then what dynamical behaviour a model is prevented from reproducing when this kind of simplifying assumption is applied to partially unknown systems.

An alternative simplification is to represent the effect of the linear pathway using a fixed time-delay in the model. Focusing on this aspect of the linear pathway and assuming that all other aspects are negligible allows for a terse model description using delayed differential equations (DDEs) [39–42]. However, it is not clear how such an assumption limits a model's ability to recapitulate the dynamics of the full system.

A third simplification is to make use of a gamma-distributed delay. This approach models the output of the linear pathway as the convolution between the input to the first pathway step and the probability density function of the gamma distribution. This convolution has been used to describe delays in a diverse set of processes, including drug uptake [43, 44], circadian clocks [45–47], population dynamics [48] and even traffic jams [49]. Furthermore, it was recently shown to be effective at simplifying specific models while still allowing them to retain the dynamical properties of the original models [47]. Similar to the fixed-delay approach, it is commonly used as a method to introduce a delay without explicit regard to what the underlying cause of that delay is. It can be derived from a chain of identical linear processes which indicates that it may be particularly relevant for linear pathways. However, an understanding of how well this simplification can represent a general linear pathway is still missing.

The problem of how to simplify non-linear pathways is even more complex. Both the dynamical contribution and the simplification of certain non-linear, sequential, pathways have been studied previously [1, 12, 32]. However, the diversity of possible non-linear pathways and their resistance to analytical exploration have left us with relatively few tools that allow for their simplification without sacrificing the ability to either describe or predict the dynamics of real pathways.

Here, we examine the dynamical effect of sequential multi-step pathways on a system and especially whether certain simplifying assumption yields models capable of reproducing those dynamical effects. We primarily focus on linear pathways—where each step is linearly dependent on the last—where we first analyse what dynamical properties a model will be unable to

reproduce when it is simplified using pathway truncation. We show how such models may be incapable of producing an output that is both as delayed and as sharply defined as the output of the full systems that they are trying to emulate. This analysis further led us to a diagnostic tool for revealing when such a model assumption has had detrimental effects.

Thereafter, we suggest the use of an alternative simplifying assumption and demonstrate its effectiveness. Rather than assuming a fixed (and truncated) pathway length, we assume a fixed rate of information propagation along a pathway of dynamic length. While similar ideas have recently been used, their effect has not to our knowledge been systematically studied before [47, 50]. We use this assumption to define a three-parameter model which can recapture the dynamics of arbitrary linear pathways with high fidelity. The assumption allows for a direct derivation of the gamma-distributed delay and it allows the model parameters to be anchored to the underlying biology. Furthermore, it outperforms the use of the reduced step approximation as well as the fixed-delay approximation and it provides a building block for an operational model inference approach [51].

Finally, we expand our analysis to non-linear pathways in the form of kinase cascades where each pathway step can become saturated. Here, we find a similar, albeit smaller, performance difference between different model assumptions when it comes to reproducing data and that the two assumptions both are reasonably good at predicting the results of novel conditions as long as they have been trained on sufficient data.

## Results

### Linear pathway truncation causes different degrees of error for different underlying distributions of reaction rates

We set out to explore the dynamical consequences of misrepresenting the number of pathway steps in a model of a linear pathway. The main aim was to understand whether and how a short (truncated) linear pathway model fails to reproduce the dynamics generated by a longer pathway.

To investigate this, we first defined a model wherein a sequence of $n$ states (which we will also refer to as steps), with concentrations $X_1, X_2, \ldots, X_n$, each activates its successor. A step $i$ in such a pathway is linear if

$$\frac{dX_i}{dt} = \alpha_i \cdot X_{i-1} - \beta_i \cdot X_i, \tag{1}$$

for some activation rate constant $\alpha_i$ and degradation/inactivation rate constant $\beta_i$. We define the entire pathway as linear if this equation holds for all steps. In this study, we specifically focus on the relationship between an input and the output of a linear pathway and not on the relative concentrations along the different pathway steps. We can thus define a set of new parameters, such that scaling is done by a single parameter, $\gamma$, and the response rate of pathway step $i$ to changes in the previous step is governed by a parameter $r_i$ (Methods). This leads to a model that describes how an $n$- step linear pathway transforms an input signal, $I(t)$, to an output, as defined by the concentration of the $n$-th step, which is given by

$$\frac{dX_1}{dt} = r_1 \cdot (\gamma \cdot I(t) - X_1), \tag{2}$$

$$\frac{dX_i}{dt} = r_i \cdot (X_{i-1} - X_i) \quad \forall i \in \{2, 3, \ldots, n\}. \tag{3}$$

Synthetic data sets were generated using Eqs 2 and 3 with different pathway lengths, $n_{data} \in \{1, 2, \ldots, 50\}$. The response rates were drawn from a log-uniform distribution, $r_i \sim 10^{\mathcal{U}(-2,1)}$, and the scaling parameters, $\gamma$, was set to one.

The effects of misrepresenting the pathway length in a model were tested on each set of synthetic data. Models with a fixed pathway step length (fixed-step models) of $n_{model} = 1, 2, \ldots, 5$, respectively, were each treated with the same input and initial conditions that were used for data generation and had the values of their parameters ($\gamma, r_i$) optimised to fit the output dynamics of that data (Methods). To best characterise how well a model could perform when $n_{model} \neq n_{data}$ we focused on studying their response to a unit impulse input, $I(t) = \delta(t)$, from an initially inactive state, $X_i(0) = 0 \;\forall i$. This input is useful since analyses of the impulse response is easily extendible to arbitrary inputs [52] (Methods). It is worth noting that the impulse input is equivalent to the pathway receiving no input at all but instead start its simulation from an initial condition where $X_1(0) = \gamma r_1$ in an otherwise fully inactive pathway. This simulates the sudden start of a reaction at $t = 0$ where $X_1$ passes on its signalling while being exponentially depleted itself and could represent the conversion of a depleting responder upon the onset of an external signal.

We analysed how well such fixed-step models reproduces the generated data and especially how the model/data fit depends on how many pathway steps were used to generate that data (Fig 1 and S1–S4 Figs). When $n_{model} \geq n_{data}$, the model can perfectly reproduce

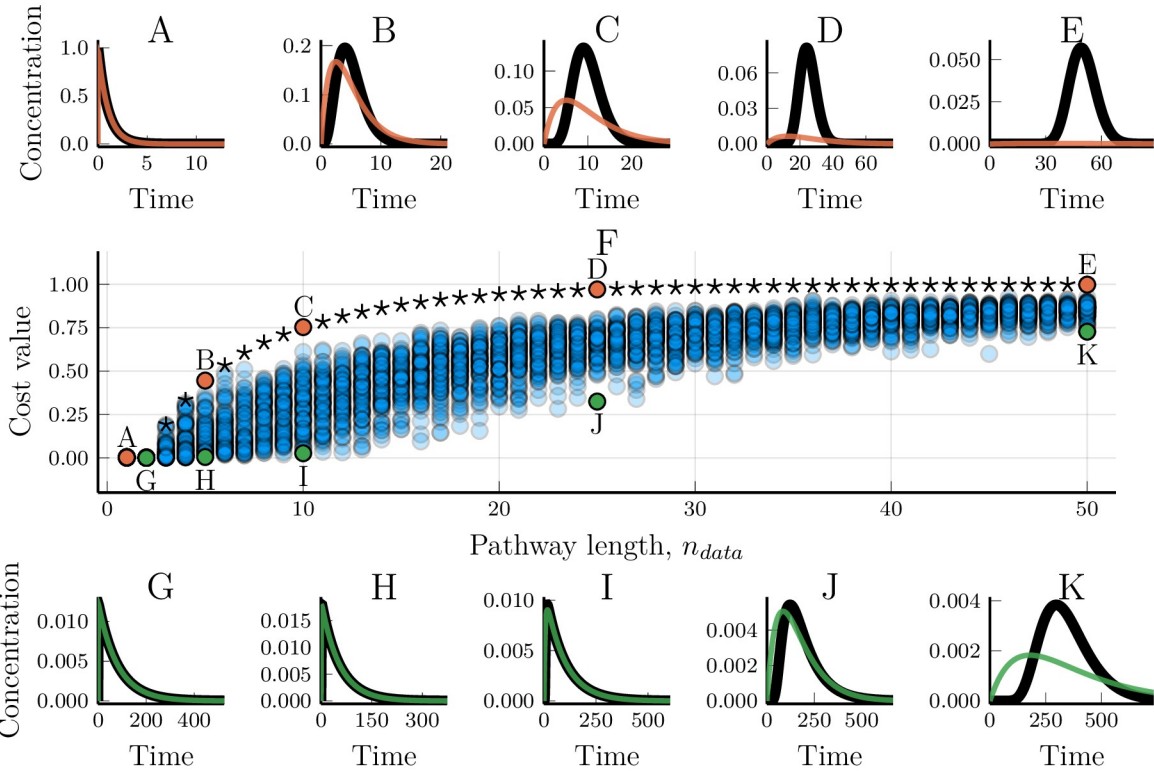

**Fig 1. Modelling linear pathways using a truncated number of pathway steps.** Two-step linear pathway models (Eqs 2 and 3) were fitted towards synthetic data. The data shown was generated by networks of step-lengths varying from 1 to 50. (A-E) The worst model/ data fits for a given length, $n_{data}$, of the model that generated the data. Black lines show the synthetic data while simulations of the fitted models are overlaid in colour. (F) The cost value for models optimised towards 5000 different sets of synthetic data. Stars are the cost values resulting from data wherein all the steps in the data-generating linear pathway have the same reaction rates, $r_i = 1 \;\forall i$. The x-axis shows the number of steps in the models that were used to generate the data. Annotations highlight the simulations that are represented in the other subplots. (G-K) Examples of the best model/data fits for different data pathway lengths, $n_{data}$.

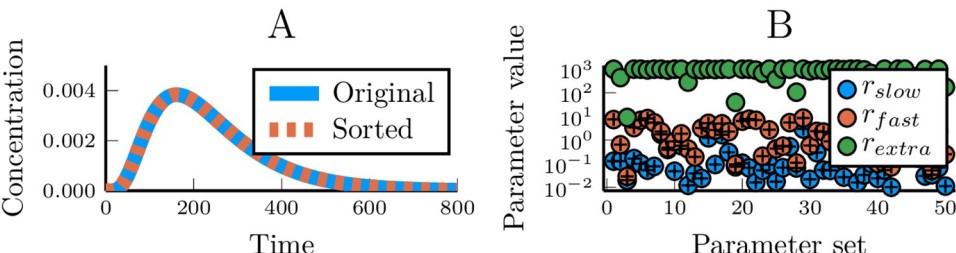

**Fig 2. Analysis of optimised parameter values for the fixed-step models.** (A) The same linear pathway model run twice, but with the order of its response rate parameters changed. (B) The optimised parameters of a model with three steps, fitted against data generated using only two steps. The model parameter values are shown as circles, and the corresponding parameter values used to generate the data are shown as crosses. The green circles represent the model parameter values of the superfluous pathway step.

the output dynamics, as expected, and provides a control for our numerical optimisation scheme (Fig 1 and S1–S4 Figs). However, despite the perfection in the input-output correspondence, the fitted model and the data-generating model will not be precisely the same. While every optimised reaction rate in the fitted model will also be found in the data-generating linear pathway (Fig 2B), they do not necessarily appear in the same order (Fig 2A). The order of the response rates along the pathway does not matter for the output [27, 28]. This means that optimising for an input-output relationship will not provide any means of correctly inferring which rate belongs to which step. Not only does the model correctly fit the data when $n_{model} = n_{data}$, but also when $n_{model} > n_{data}$ (Fig 1 and S2–S4 Figs). During the optimisation procedure, 'additional' rates become fast enough to 'instantaneously' pass information from the previous step to the next (Fig 2B), confirming that fast steps are less dynamically relevant than slow steps [1, 27]. When the model has fewer steps than the linear pathway that was used to generate the data ($n_{model} < n_{data}$) it may no longer be possible to find a good fit. Unsurprisingly, the model/data mismatch increases with the length of the pathway that generated the data and decreases with the length of the model used to fit the data (Fig 1 and S1–S4 Figs).

There is a high variability in the ability of a truncated model to fit the output dynamics (Fig 1 and S1–S4 Figs). While small models cannot in general represent arbitrary linear pathways, in some cases they do perform well. For example, a two-step model can be good at reproducing the dynamics of even a 10-step linear pathway (Fig 1J), and a five-step model can accurately describe the dynamics of some 25-step pathways (S4 Fig). However, the optimised model performance decreases with the homogeneity of the reaction rates of the data-generating pathway (Fig 3). When all the response rates of the data-generating pathway are the same, the ability of models to fit the data quickly decreases with the length of the pathway (Fig 3, cf. Fig 1A–1E, stars in Fig 1F). Conversely, when the response rates are highly heterogeneous, even a heavily truncated model is able to fit data from a long pathway (Fig 3), again indicating the different contributions of fast and slow steps to the resulting dynamics. Here, we see evidence that the relative distribution of response rates in the data-generating pathway affects the truncated model's ability to fit data, next we will more rigorously examine why.

## Detrimental truncation of models lead to unsharp responses but can be detected by characteristic parameter values

While pathway length and response rate homogeneity are key determinants for whether the truncation of a linear pathway reduces a model's accuracy, these features may often be unknown

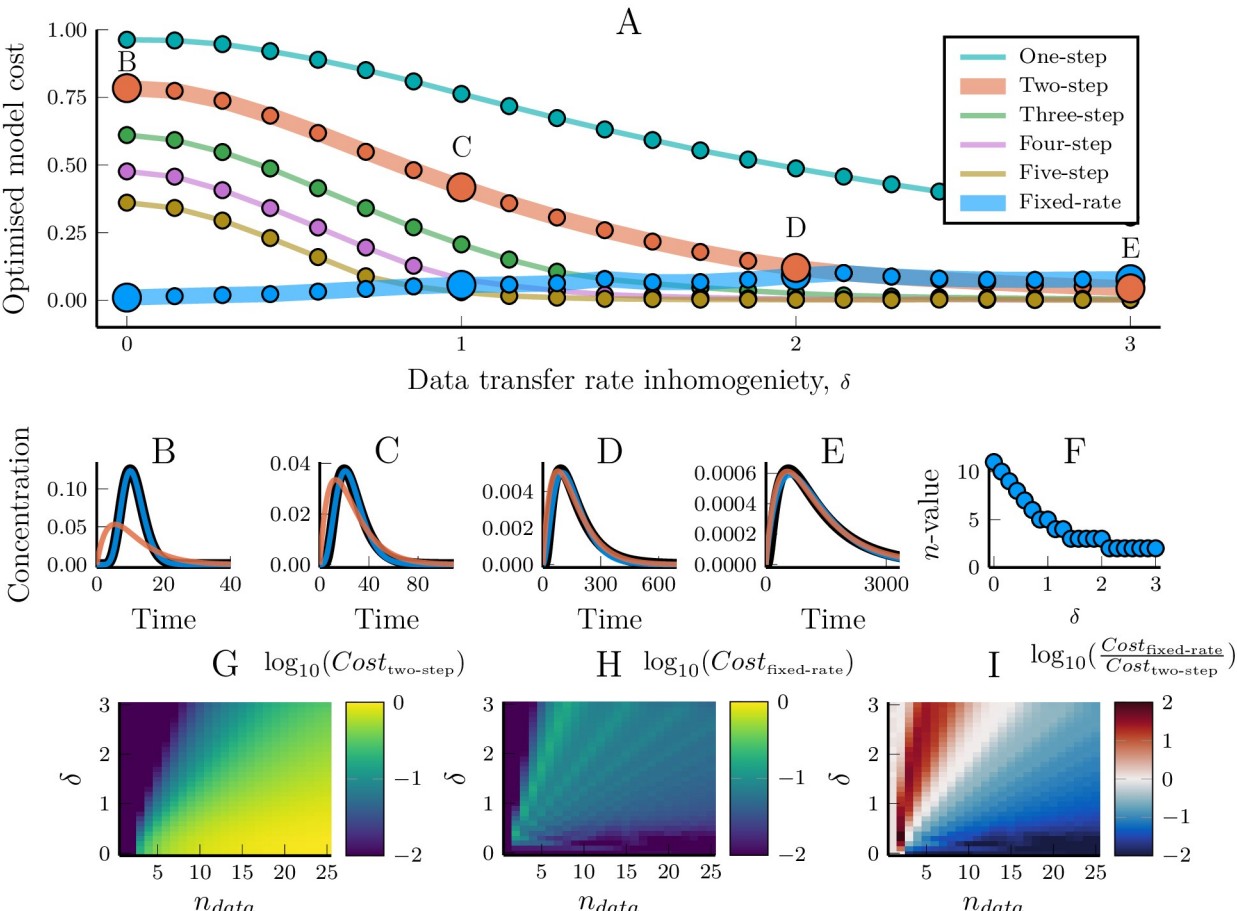

**Fig 3. Response rate homogeneity of a linear pathway affect how well models can reproduce their dynamics.** Models were fitted towards data that was generated with different levels of response rate inhomogeneity. The synthetic data was generated using linear pathway models (Eqs 2 and 3) with parameters $\gamma = 1$ and $r_i = 10^{\delta \cdot \left(2 \cdot \frac{i}{n_{data}} - 1\right)} \; \forall \, i \in \{0, 1, \ldots, n_{data} - 1\}$ where $\delta$ is a parameter that governs the inhomogeneity of the rate parameters. For $\delta = 3$, the rate parameters were thus logarithmically spread from 0.001 to 1000. (A) The cost values (lower is better) of models fitted to an 11-step linear pathway, plotted against the inhomogeneity parameter, $\delta$, used to generate the synthetic data. (B-E) Samples of the model/data fit for the optimised parameter sets, as marked in (A). The fixed-rate model (Eqs 8 and 9) is compared with the two-step model (Eqs 2 and 3) since they have the same number of free parameters. (F) The optimised value for the fixed-rate model parameter $n$ compared with the inhomogeneity of the response rates ($n_{data} = 11$). (G) and (H) The cost values for the two-step model and the fixed-rate model, respectively, when fitted to pathways of different lengths and degrees of inhomogeneity. (I) The ratio of the cost values for the fixed-rate and the two-step models. Blue indicates that the fixed-rate model did better while red indicates that the two-step model did better. White indicates that the two models performed equally well.

from experiments. Hence, it would be useful to find characteristics of the (simplified) model to quantitatively identify when it is performing badly and how that can be detected.

In order to characterise the response curve, we define its delay, duration and sharpness. For signal delay and duration, we adapt definitions and linear-pathway specific relations described in Heinrich et al. [12]. The delay, $\tau$, is defined as the average time it takes for an input to activate the output which can be expressed as

$$\tau \equiv \frac{\int_0^\infty t X_n(t) dt}{\int_0^\infty X_n(t) dt} = \sum_{i=1}^{n} \frac{1}{r_i}, \tag{4}$$

where the second equality holds for linear pathways. This definition is similar to the mean of a

statistical distribution. The signal duration is defined in a way that is similar to a standard deviation of a distribution, given by

$$\sigma \equiv \sqrt{\frac{\int_0^\infty t^2 X_n(t)dt}{\int_0^\infty X_n(t)dt} - \tau^2} = \sqrt{\sum_{i=1}^n \frac{1}{r_i^2}}, \tag{5}$$

where, again, the second equality holds for linear pathways. We also define the sharpness of the output curve, $s \equiv \frac{\tau}{\sigma}$, as the reciprocal of a normalised signal duration. By using the Cauchy-Schwarz inequality and that the sum of squares is smaller than or equal to the square of a sum we get that

$$\frac{\tau}{\sqrt{n}} \leq \sigma \leq \tau \tag{6}$$

for linear pathways. This means that for any given signal delay, the signal duration is limited from both above and below. The inequality is equivalent to

$$1 \leq s \leq \sqrt{n} \tag{7}$$

which shows that the sharpness of the output curve is bounded by the square root of the pathway length. This means that when a linear pathway is simplified by reducing the number of pathway steps, the resulting model may become incapable of producing as sharp an output curve as the original system.

Here, we see why a short model is only occasionally able to capture the dynamics of a long one and why the homogeneity of the response rates matter. It is because even though the *maximal* sharpness of a longer pathway is greater than what a truncated model can emulate, the *actual* sharpness might not be. Data from homogeneous pathways (where $r_i = r \, \forall i$) is hard for a truncated model to fit since the actual sharpness of the data matches its maximal sharpness, $s_{data} = \sqrt{n_{data}}$. The actual sharpness of the data curve, $s_{data}$, decreases as the response rates become more heterogeneous and it becomes easier for a truncated model to reproduce the data. The linear pathways which are the most effectively truncated are those where all but one step is infinitely fast since this results in the minimal sharpness $s_{data} = 1$. Such data can be reproduced by any fixed-step model of length one or more.

A consequence of this is that a detrimentally truncated model creates a clear signature in the optimised parameter values. While one could try to detect detrimental truncation by inspecting the maximal sharpness of the model and comparing to the actual sharpness of the data this only works for an impulse-like input and it requires that the direct output of the linear pathway is experimentally observable. A more reliable method to detect detrimental truncation comes from how when the data has a sharper peak than a fixed-step model is able to reproduce, the fixed-step model will optimise to have its maximal sharpness, $s_{model} \rightarrow \sqrt{n_{model}}$ (Fig 4). This limit is achieved when all the model's response rate parameters have the same value (since $\frac{\tau}{\sigma} = \sqrt{n}$ when $r_i = r \, \forall i$). We thus see that a fixed-step model that has been simplified to the point that it can no longer capture the dynamics of the data will optimise to have homogeneous rate parameters. Detrimental pathway truncation can thus be detected by inspection of the optimised parameters of a model. If a chain of steps in a model has the same response (or degradation) rates, this indicates that a linear pathway might have been truncated to the detriment of the model/data fit. Importantly, while this analysis is based on the assumption of an impulse input to the model, it holds for essentially all inputs. This is because we can expect the optimal model parameters in just about any relevant scenario to be those that

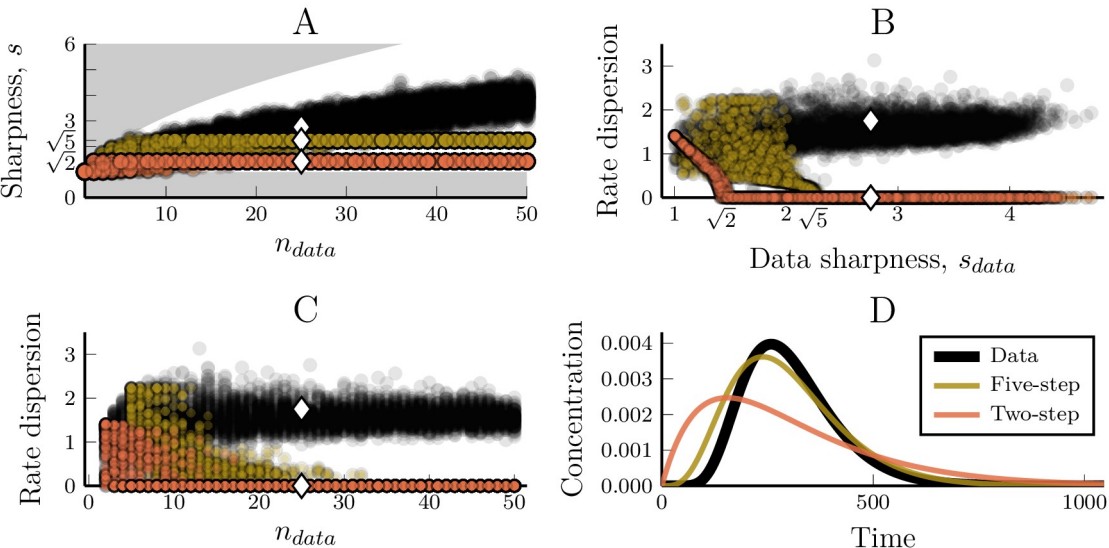

**Fig 4. The fixed-step model has a maximal peak sharpness that can both prohibit data recapture and provide a diagnostic tool for when this occurs.** (A) The sharpness of the output peaks for the synthetic data and fitted two and five-step models. The shaded regions show the sharpness bounds for the data ($1 \leq s_{data} \leq \sqrt{n_{data}}$) and the white diamonds highlight the parameter sets used for the example trajectory in (D). (B) The dimensionless dispersion of rate parameters of the data and fitted models compared to the sharpness of the data. We define the dispersion to be the standard deviation over the mean of the rate parameters of a single model instance. Zero dispersion means that all the rates equal one another. (C) The rate parameter dispersion compared to the length of the pathway that generated the data. (D) The time-course of the model instances marked as diamonds in (A-C).

enables the closest possible match between the transfer functions of the model and the data. Since these transfer functions are input-independent so is this diagnostic.

## An alternative assumption for model simplification improves predictability of pathway output dynamics

An alternative approach for parameter reduction is to assume that every pathway step has the same response rate ($r_i = r \ \forall i$), and to treat the number of pathway steps, $n$, as a free parameter. With this 'fixed-rate' assumption we can represent a linear pathway with the set of equations

$$\frac{dX_1}{dt} = r \cdot (\gamma \cdot I(t) - X_1), \tag{8}$$

$$\frac{dX_i}{dt} = r \cdot (X_{i-1} - X_i) \quad \forall i \in \{2, 3, \ldots, n\}. \tag{9}$$

This simplified model has only three free parameters: $\gamma$ for scaling, $n$ for pathway length, and $r$ for the response rates of the pathway steps.

While the assumption of homogeneous reaction rates is natural when the same process is repeated multiple times, such as a molecular motor walking along a microtubule, or for the assembly of monomers into a polymer, it is not true for most pathways. We analysed the effectiveness of the fixed-rate model by individually fitting its three parameters towards each of the synthetic data sets used above. The resulting model/data fits show that the fixed-rate assumption is indeed well suited for modelling linear pathways, even when the reaction rates of the data-generating network are highly heterogeneous (Figs 5 and 3). The relative effectiveness of the fixed-rate and the fixed-step model depends on the nature of the data itself. For very short or highly heterogeneous pathways, the fixed-step assumption yields better results (Fig 3I).

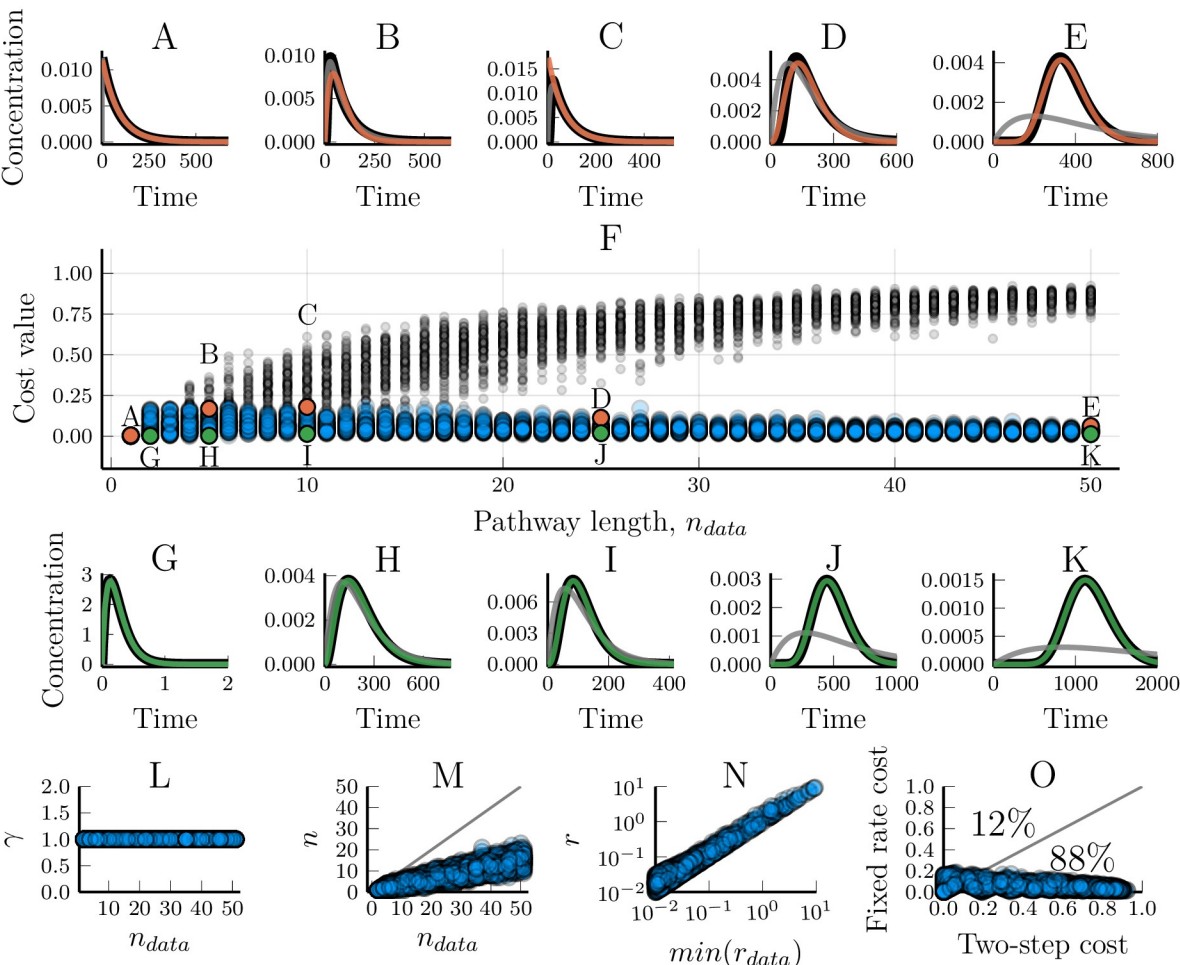

**Fig 5. Modelling linear pathways using the fixed-rate assumption for simplification.** The fixed-rate model (Eqs 8 and 9) was fitted towards synthetic data generated by networks of step-lengths varying from 1 to 50. The data sets are the same as those used for Fig 1. (A-E) Examples of the worst model/data fits for the fixed-rate model. Orange lines show simulations of the fitted model and the black line shows the synthetic data. The grey line is the corresponding fit using the two-step truncated model on the same data set. (F) The cost value for models optimised towards 5000 different sets of synthetic data. The x-axis shows the number of steps in the model which were used to generate the data. Blue circles are cost values for the fixed-rate model while grey dots are the cost values for a two-step truncated model. The parameter sets used in figures A-E and G-K are marked accordingly. (G-K) Examples of the best model/data fits. The fixed-rate model (green lines) almost completely matches the data (black lines). (L) The scaling parameter, $\gamma$, is accurately identified as 1 in all optimisations. (M) The optimised values of $n$ for different number of steps in the pathway underlying the data. (N) A comparison between the optimised value of $r$ and the smallest rate constant of the model that generated the data. (O) A comparison of the cost values when using either the fixed-rate model or a two-step truncated model to fit the same data. Each circle represents a single synthetic data set. Percentages indicate how many of the data sets had a higher (worse) cost value for the respective models.

That said, the fixed-rate model outperformed the two-step model, which has the same amount of free parameters, in a clear majority of the cases we examined (Fig 5). This holds true for a wide range of different inputs to the linear pathway (Fig 6). Strikingly, the fixed-rate model was almost inseperable from the output of the original model in most cases (blue and black lines in Fig 6). It should also be noted that the fixed-rate model is reliably good (cost values less than 0.2), even in the cases where the fixed-step model is better (Fig 3G–3I). In contrast, the fixed-step model can become very bad at representing the data (with cost values above 0.9) when the data peak is sharper than the model can reproduce. Increasing the number of steps in the fixed-step model does increase its ability to fit the data (e.g. Fig 4, S1–S4 Figs). However,

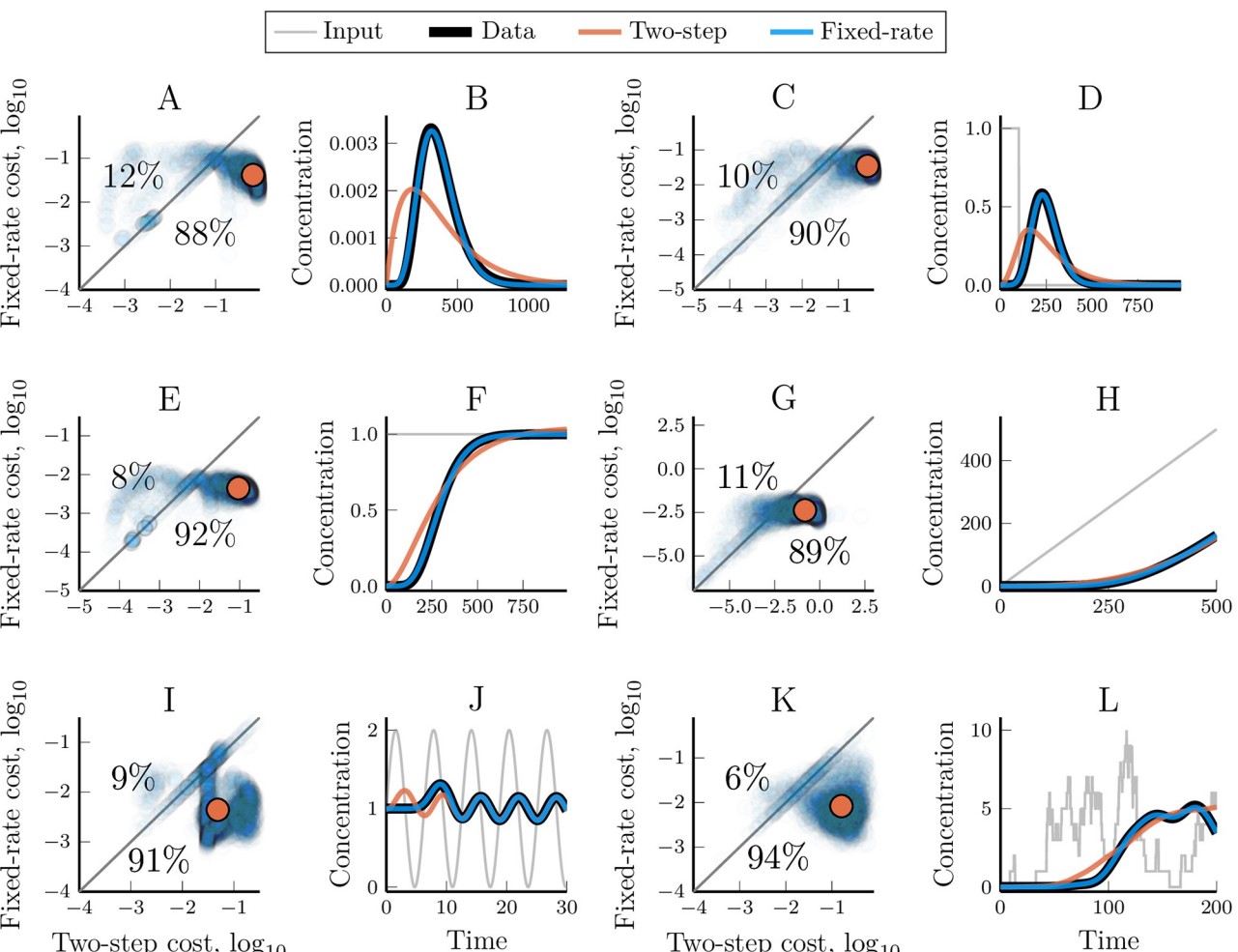

**Fig 6. Comparing the ability of the fixed-rate and the two-step model to reproduce the dynamics of linear pathways that respond to different inputs.** Each pair of figures (A,B; C,D; . . .; K,L) demonstrates the performance of the two models for the different model inputs: impulse, piecewise constant, ramp, step, wave and noisy auto-activator, respectively (see Table 1). Figures A, C, E, G, I and K compares the optimised cost values (lower is better) for the fixed-rate model and the two-step model for each synthetic data set (5000 per input). Every data set is represented by a low-opacity dot; colour saturation thus indicates the density of similar values. Percentages indicate how often one model had a worse cost than the other. The orange dot shows the geometric median of the cost values for the different data sets. Figures B, D, F, H, J and L shows the model and data dynamics for the median data set of the corresponding input. The two-step model was chosen for the comparison since it has the same number of free parameters as the fixed-rate model.

this adds more parameters while it does not change the fundamental limitation that when the data is too sharp for the model there is no limit to how bad the fit can become. The fixed-rate model was usually better in our test cases, even compared to longer fixed-step models (S6 Fig). It is thus highly effective at simplifying a linear pathway, and the performance only decrease slightly when the underlying pathway has heterogeneous reaction rates (Fig 3). More importantly, the fixed-rate model proved *reliably* good when its more common counterpart did not.

To verify the applicability of the fixed-rate model beyond our synthetic data, we compared its performance with that of the two-step model when fitting real data. For that, we used data from the immune response of *Arabidopsis thaliana*. There, a set of different surface receptors respond to different danger-associated inputs and then, through a multi-step pathway, trigger an output of reactive oxygen species (ROS) [23]. The ROS response is transient even though the input is persistent and this is hypothesised to be due to input-induced degradation of the receptor. We

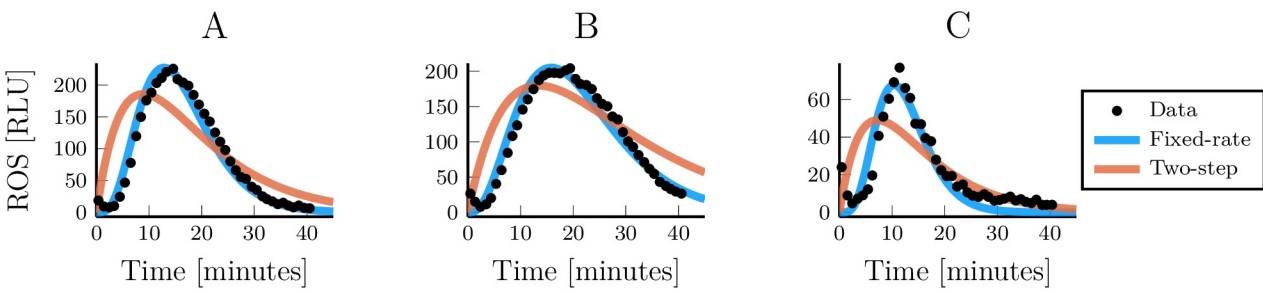

**Fig 7. Simplified linear pathway models fitted to real data from different immune responses of *Arabidopsis thaliana*.** (A-C) Measured and simulated reactive oxygen response to immune elicitations by flg22, elf18 and AtPep1, respectively. Data extracted from Monaghan et al. [53].

can interpret this as an initially inactive pathway where only the first step (the inactive receptor) has a non-zero concentration. At time $t = 0$, the receptors start binding to the input and keeps on producing signalling until they are degraded. This case is analogous to the impulse input that we have used in the paper. The multi-step pathway leading to the release of ROS is not perfectly characterised and we do not know whether or not it is linear. We do, however, know that for one of these inputs, flagellin22 (flg22), there is a sequence of at least five reactions that occur before the production of ROS is triggered [23]. Fitting the models to this data clearly shows that the fixed-rate model is better able to capture the experimentally observed dynamics (Fig 7).

## The fixed-rate model outperforms a fixed-delay model

A main limitation for the truncated models is that they perform badly in capturing the delay of a signal output. Rather than introducing multiple pathway steps in a model in order for it to capture a time-delay, such time-delays can be introduced explicitly. We next aimed to compare such an approach of modelling linear pathways to the use of the fixed-step and the fixed-rate formulations. In order to make a fair comparison, we defined a DDE with three free parameters. In this 'fixed-delay' model, the pathway step that we consider an output, $X$, responds to an input at a rate governed by $r$, with a fixed, singular, time delay, $t_{delay}$. Similarly to the other models, a scaling parameter $\gamma$ is also defined.

$$\frac{dX(t)}{dt} = r \cdot (\gamma \cdot I(t - t_{delay}) - X(t)). \tag{10}$$

We optimised the fixed delay model (Eq 10) towards the same data used for the previous models and compared their performance (Fig 8). In most cases, this model performed better than the two-step truncated model but worse than the fixed-rate model. The fixed-delay model is able to adjust its signal delay (Eq 4) without affecting the signal duration (Eq 5). This allows it to fit the delay, duration, and thus, sharpness of the data well. However, it fails to fit the symmetry of the output curve around its mean delay (skewness if we continue the distribution analogy). The model output starts very abruptly at $t = t_{delay}$ and its response to sudden input changes lacks the smoothness that is seen in the data. The fixed-rate model, on the other hand, can achieve the correct time-delay and sharpness while also accurately smoothing out the signalling over time.

## The fixed-rate assumption leads to a gamma distributed delay model

The impulse response function for the fixed-rate model is given by

$$g(t) = \frac{\gamma \cdot r^n \cdot t^{n-1} \cdot e^{-r \cdot t}}{\Gamma(n)}, \tag{11}$$

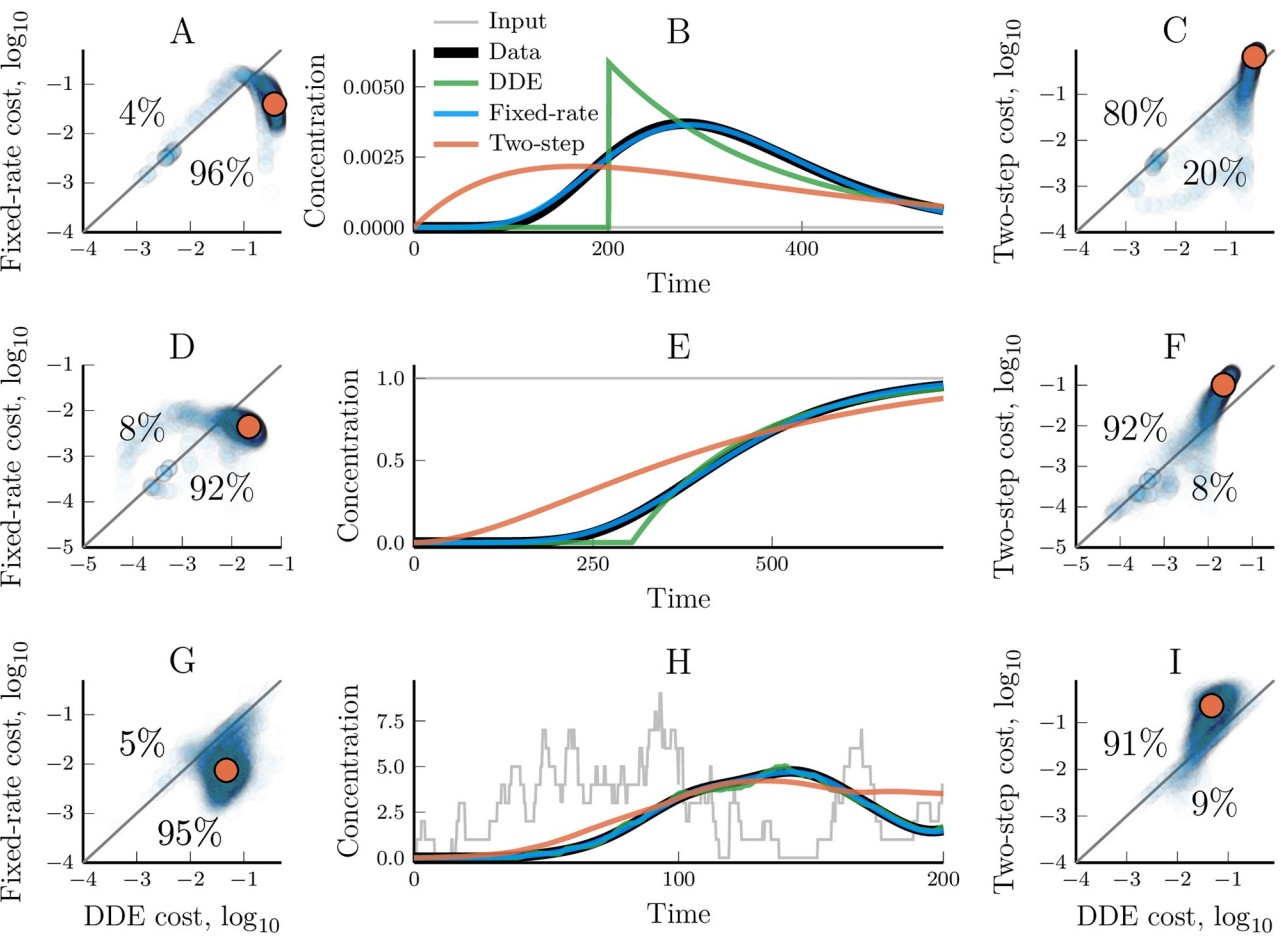

**Fig 8. Modelling of linear pathways using a fixed delay DDE model (Eq 10) compared to the two-step and the fixed-rate models.** (A) A cost value comparison of the fixed-rate model and the DDE model for every data set generated with an impulse input (Table 1, 5, 000 data sets). Every synthetic data set is represented with a low opacity dot; colour saturation thus indicate a high density of similar values. The orange dot highlights the geometric median and that median data set is used as an example in figure (B). (B) An example time trajectory where the fixed-rate, two-step and DDE models have all been optimised to reproduce a synthetic data set. (C) A cost value comparison, similar to (A), between the two-step model and the DDE model. The orange dot shows the cost of the trajectory displayed in (B). (D-F and G-I) Repeats of (A-C) but using a step input and a noisy input, respectively (Table 1). The synthetic data was generated with pathway lengths, $n_{data}$, uniformly distributed between 1 and 50.

as shown in Methods. Here, $t$ is time, $\Gamma$ is the gamma function [54] and, just like in the fixed-rate model, $\gamma$ is a scaling factor, $n$ relates to the pathway length and $r$ to the response rates of the steps along the pathway. For the impulse input, the fixed-rate model is easily simulated by solving $X_n(t) = g(t)$. Similarly terse analytical solutions to the fixed-rate model for other, specific, inputs can be found in [27]. But the impulse response function is of particular importance since it can be used to derive an expression which is applicable to any kind of input. This is done through a convolution and because $g(t)$ is really identical to the probability density function of the gamma distribution (scaled with $\gamma$) the result is called a gamma-distributed delay,

$$X_n(t) = g(t) * I(t) = \int_0^t g(t - t') \cdot I(t')dt'. \tag{12}$$

While the fixed-rate model is by its very structure limited to integer-valued $n$, this 'gamma model' is not and allowing real-valued $n$ increases the model's ability to fit data. To test this,

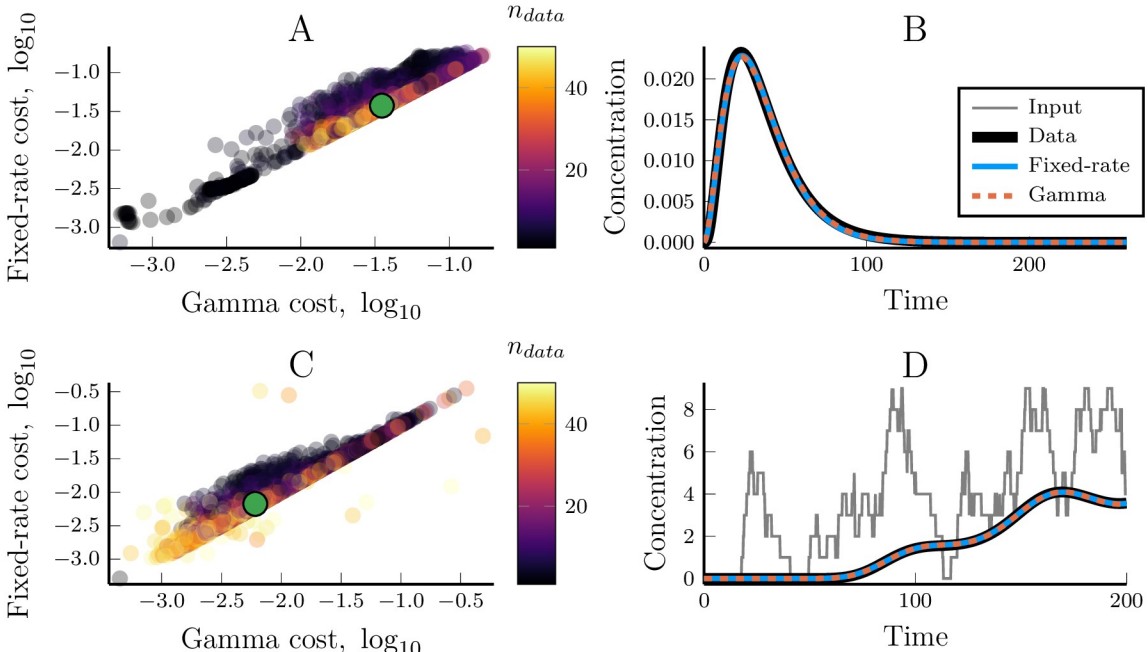

**Fig 9. Allowing a real-valued pathway length parameter, *n*, increases the gamma model's ability to recapitulate dynamics from arbitrary linear pathways.** (A) The optimised cost values for the fixed-rate and the gamma models when they are both optimised towards the same data sets in which the pathway receives an impulse input (Table 1, 5000 data sets). The colour of the dots indicate the length of the linear pathway that was used to generate the synthetic data set. (B) Demonstrating the model performances for the median data set, as denoted by a green dot in (A). (C, D) similar to (A) and (B) but using the generalised gamma model (Eq 12) and a noisy input (Table 1).

we optimised the gamma model and the fixed-rate model towards the same synthetic data sets. The gamma model, with $n \in [1, \infty)$, improved performance compared to the fixed-rate model in nearly all cases (Fig 9, cf. S5 Fig, [27]). In most cases, the two models were approximately equivalent but the performance difference became apparent in some cases, especially when $n_{data}$ was small.

Due to the close connection between the gamma model and the gamma distribution the model's signal delay and duration is directly given by the statistics of the distribution. The mean of the gamma distribution identifies the signal delay, $\tau = \frac{n}{r}$, standard deviation identifies the signal duration and is given by $\sigma = \frac{\sqrt{n}}{r}$. This also means that the sharpness of the gamma model (as well as of the fixed-rate model) is given by $s = \frac{\tau}{\sigma} = \sqrt{n}$. This means that the gamma model is always able to fit its signal sharpness exactly to that of the data. The fixed-rate model, on the other hand, is only approximately able to fit the sharpness and the relative error is largest for data with low sharpness due to the limitation of an integer-valued *n*.

## Identifiability of biological parameters using the fixed-rate assumption

It is of interest to analyse how well a fixed-rate assumption approach performs when it comes to identifying the values of the parameters in the underlying biological pathway. When a pathway is using the same reaction rates for all pathway steps, the fixed-rate assumption is exact, and the optimal model parameters truly reflect the underlying parameters of the data-generating system. However, the model performs well even when representing a pathway with heterogeneous reaction rates, and in this case, the connection between the model parameters and the biological system is less clear.

The $\gamma$ parameter is a simple scaling parameter. The model parameter can be considered a reflection of the total scaling that a linear pathway performs on a signal before that signal reaches the output (Methods, Eq 20). Unlike for the fixed-step model, the optimised values of $\gamma$ for the gamma and the fixed-rate models were all very close to the true value, one (Fig 5L). All of these models are capable of ensuring that the signalling is properly scaled between the input and the output. However, since the fixed-step model is unable to correctly time its output, the optimisation scheme will sometimes lead to the use of the scaling parameter to mitigate the cost that this timing discrepancy creates. Since the fixed-rate assumption leads to models with much better control over timing, this never became an issue during our study, and those models always identified the correct value for the scaling parameter.

The $n$ parameter represents the number of steps in the approximated pathway but since we relaxed the demands that all rates are equal the connection between $n$ and the number of steps in the biological (synthetic) data is not exact. Nevertheless, a precise relationship between the optimal pathway length parameter, $n_{optimal}$, and the parameters of the approximated pathway can be found. This can be done if we assume that the model fits the data optimally when the two share both signal delay and duration ($\tau_{model} = \tau_{data}$ and $\sigma_{model} = \sigma_{data}$). Such optimality is only approximately possible for the fixed-rate model but it is strictly possible for the gamma model. In this case, we find that $n_{optimal}$ can be determined from the rates, $r_i$, of all the pathway steps of the data

$$n_{optimal} = \frac{\tau_{model}^2}{\sigma_{model}^2} = \frac{\tau_{data}^2}{\sigma_{data}^2} = \frac{\left(\sum_i^{n_{data}} \frac{1}{r_i}\right)^2}{\sum_i^{n_{data}} \frac{1}{r_i^2}} . \tag{13}$$

Here, we see that if the rates of the underlying linear pathway are known, or if the gamma model is used to simplify a known linear pathway model, the optimal value of $n$ is easily obtained. This equation holds with good precision for the numerically optimised parameters of this paper (Pearson correlation for the gamma and the fixed-rate models using the impulse input, $\rho_{gamma} = 0.999$ and $\rho_{fr} = 0.997$, respectively). It should, however, be noted that the slight imperfection is not merely due to failures of our numerical optimisation scheme but mostly because our cost function was not perfectly in line with the assumption above. Even so, the relation is still informative and by using the Cauchy-Schwarz inequality it further tells us that

$$n_{optimal} \le n_{data}. \tag{14}$$

This means that while the value of the optimised $n$ parameter does not give the precise length of the underlying pathway, it does give a lower bound (Fig 5M). Again, equality occurs when all the response rates of the underlying pathway have the same values.

The parameter $r$ of the fixed-rate model is related to the rates at which information is being passed along the linear pathway. By again assuming that the optimal model fit occurs when the signal delay and duration of the model and the data are equal, we get

$$r_{optimal} = \frac{\tau_{model}}{\sigma_{model}^2} = \frac{\tau_{data}}{\sigma_{data}^2} = \frac{\sum_i^{n_{data}} \frac{1}{r_i}}{\sum_i^{n_{data}} \frac{1}{r_i^2}} . \tag{15}$$

Where, $r_{optimal}$, is the optimal $r$ parameter value of the gamma model and $r_i$ are the response

rates along the approximated linear pathway. Like with $n_{optimal}$, this $r_{optimal}$ corresponds very closely to the optimised $r$ parameters of this paper ($\rho_{gamma}$ = 0.999, $\rho_{fr}$ = 0.996). While this relationship can be useful, it provides little in terms of intuition. A more intuitive, albeit less precise, heuristic is that the parameter $r$ approximately identifies the slowest part of the pathway that is being simplified. This is because the slowest steps provide the greatest contributions to the overall dynamics [12]. This heuristic turns exact both when all the rates of the data-generating pathway are the same as well as when they tend to infinite heterogeneity. For the synthetic data of this paper, which is near neither of these limits, the optimised $r$ parameter still corresponds well with—and only slightly overestimates—the slowest reaction rate of the data (Fig 5N, $\rho_{gamma}$ = 0.993, $\rho_{fr}$ = 0.989).

Counter-intuitively, the connection between parameters and observables in the underlying data is a little bit weaker for the fixed-rate model which does not assume that the number of pathway steps can be a non-integer. This is because it may not be possible to simultaneously have both the delay and the duration of the fixed-rate model and the data be equal. The above arguments still apply but only approximately. Adapted versions of Eqs 13 and 15 could still yield good parameters but they may not be optimal.

Altogether, the fixed-rate assumption allows for a relatively close connection between a model's parameters and the observables that they represent. The optimised parameters can thus strongly indicate the properties of the linear pathway under study.

## The fixed-rate assumption is applicable to non-linear pathways

Many multi-step, unbranched, pathways are in fact not linear for all inputs. The linear approximation is often accurate for low inputs but reaction rates tend to saturate at sufficiently high inputs. It would, therefore, be useful to be able to efficiently simplify also non-linear multi-step reactions.

We, again, generated synthetic data, but this time from a non-linear kinase cascade model equivalent to that of Heinrich et al. [12]. Each step, $i$, in this model consists of the conversion back and forth between an active ($X_i^{on}$) and an inactive ($X_i^{off}$) kinase. The total amount, $X_i^{tot} \equiv X_i^{on} + X_i^{off}$, is conserved and is treated as a parameter. The conversion from inactive to active kinase is proportional to the activity of the previous kinase, $X_{i-1}^{on}$, and the availability of $X_i^{off}$, with a proportionality constant $\tilde{\alpha}_i$. The activation of the first step is proportional to the input, $I(t)$, and the activity of the last step, $X_n^{on}(t)$, is considered the model's output. The inactivation of all steps is proportional to their activity $X_i^{on}$, with a proportionality constant $\beta_i$. Similar to how we redefined the parameters of the linear model to increase their orthogonality, we here define the dimensionless parameter $\alpha_i \equiv \frac{\tilde{\alpha}_i \cdot X_i^{tot}}{\beta_i}$ and express the model as

$$\frac{dX_1^{on}}{dt} = \beta_1 \cdot \left( \alpha_1 \cdot I(t) \cdot \left( 1 - \frac{X_1^{on}}{X_1^{tot}} \right) - X_1^{on} \right) \tag{16}$$

$$\frac{dX_i^{on}}{dt} = \beta_i \cdot \left( \alpha_i \cdot X_{i-1}^{on} \cdot \left( 1 - \frac{X_i^{on}}{X_i^{tot}} \right) - X_i^{on} \right) \quad \forall i \in \{2, \ldots, n\} . \tag{17}$$

Data sets were generated by simulating the activity of the last step, $X_n^{on}$, using random parameters where $\alpha_i \sim 10^{\mathcal{U}(-1,2)}$, $\beta_i \sim 10^{\mathcal{U}(-1,2)}$, $X_i^{tot} \sim 10^{\mathcal{U}(-1,1)}$, and $n \sim \{1, \ldots, 20\}$. For sufficiently small inputs or high values of $X_i^{tot}$, this cascade model will be linear since each step will only be weakly activated such that $1 - \frac{X_i^{on}}{X_i^{tot}} \approx 1$. The cascade model is then dynamically equivalent to its linear counterpart in the study.

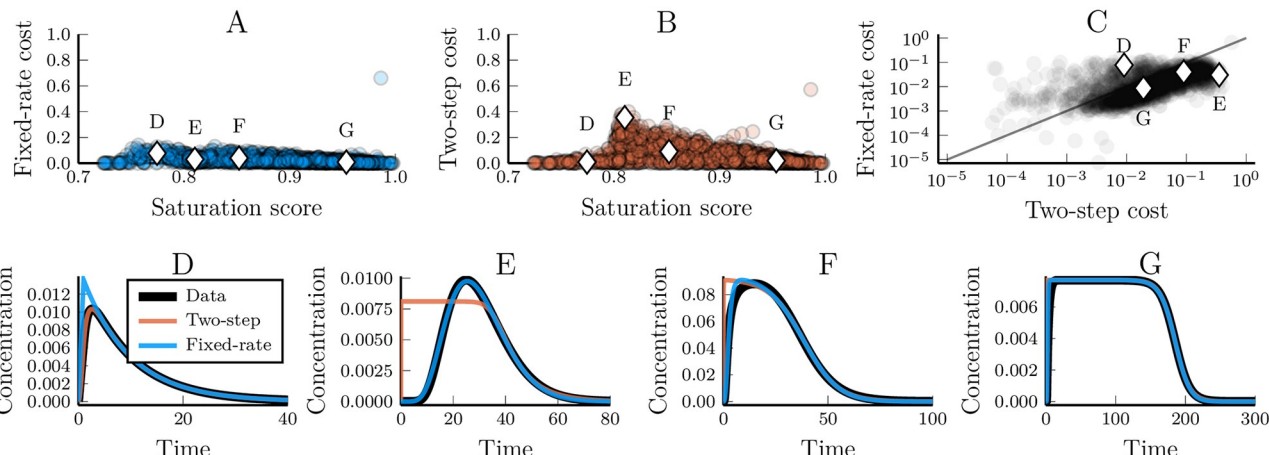

**Fig 10. The descriptive power of simplified non-linear kinase cascade models depends on the saturation level of the approximated system.** (A and B) The fixed-rate and two-step assumptions respective dependency on the saturation level of the data it tries to emulate. The saturation is measured by a quantity that depends on the shape of the final peak (Methods). Linear systems receive a saturation score of between 0.7 (for one-step systems) to 0.83. A score of 1 means that the peak is completely square. Circles represent a single data/model pair and diamonds highlight example curves in (D-G). (C) Comparing the cost values of the data-fits for models using the fixed-rate or two-step assumption.

Simplifications of such cascades using the fixed-rate assumption often yields greater descriptive power than those using a two-step assumption. We simplified the cascade model described in Eqs 16 and 17 using either a fixed two-step assumption ($n = 2$, independent $\alpha_i \beta_i$ and $X_i^{tot}$) or a fixed-rate assumption ($n$ is a free parameter but each step is the same: $\alpha_i = \alpha_j$, $\beta_i = \beta_j$ and $X_i^{tot} = X_j^{tot} \; \forall i, j$). These models were fitted towards reproducing the synthetic data. The fixed-rate cascade model yielded better fits with data than the two-step cascade model in a majority of cases (Fig 10). This holds despite the two-step model having more free parameters than the fixed-step model (6 compared to 4). Also, the fixed-rate model describes the data well regardless of how saturated that data is (Fig 10A). The two-step model, on the other hand, performs best when either the underlying pathway really is short or when the data is highly saturated (Fig 10B).

In the case of linear pathways, any simplification will have predictive power as long as they can emulate the transfer function of the original system. For non-linear models, however, we have no such guarantees. A model that fits the data perfectly under one condition may nevertheless give false predictions for how the system would react to others. We, therefore, analysed the predictive power of the two model simplifications by fitting them to data from one input and then measuring the model/data mismatch that ensued from applying a differently scaled input (Fig 11A–11D). The result shows that while certain dynamics aspects, such as maximal peak height, is often unchanged in both the data-generating model and the simplified model other aspects such as signal duration are often poorly predicted. However, predictiveness was dramatically improved for both models when we instead fitted them to data from three inputs of different scaling ($10^{-3}$, 1 and $10^{-3}$)(Fig 11E–11H). In this case, we saw very little difference in the effectiveness between the fixed-rate and the two-step assumptions except that the fixed-rate assumption requires fewer parameters. Finally, it is worth pointing out that both models performed reasonably well in this regard and that while the fixed-rate model does not always outperform the fixed-step model, it still performed reliably well in our test cases in a way that the fixed step model did not.

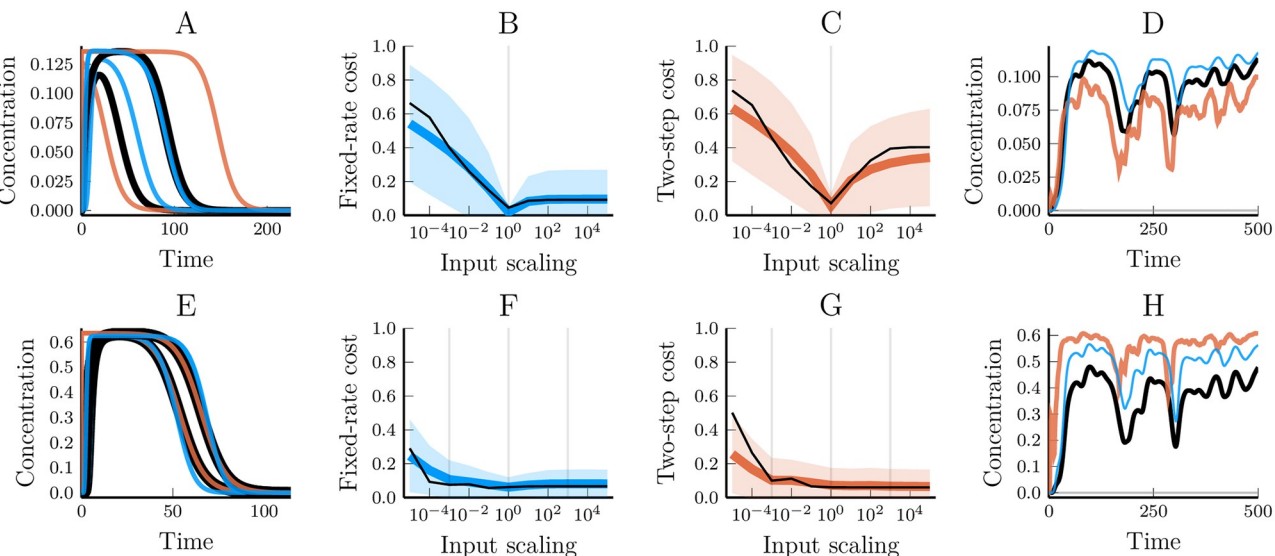

**Fig 11. The predictive power of simplified non-linear kinase cascade models when fitted towards a single vs. three different inputs.** (A) Fixed-rate model (blue), two-step model (orange) and synthetic data (black) simulated for three different pulse inputs. The inputs were pulses of concentrations $10^{-3}$, 1, and $10^3$, respectively–lasting one time unit. The models were only fitted to the unscaled input. (B and C) The cost values of the fixed-rate and two-step models, respectively, for different input scalings. The coloured line shows the mean cost for 2000 fitted models, the shaded region show the standard deviation and the black line highlights the model used as an example in (A) and (D). (D) Applying a noisy input to models that were trained on a single pulse input. Colours like in (A). The generated noise input was rescaled by a factor $2 \cdot 10^{-6}$ to avoid the uninteresting case of full and continuous saturation. (E-H) Repeating the above figures but with models trained on data from pulse-inputs with three different scalings, $10^{-3}$, 1, and $10^3$. The noise input rescale factor in (H) was $10^{-7}$.

## Discussion

Biology is full of multi-step pathways where each step is (at least approximately) linearly dependent on the previous step. Transcription, translation, kinase cascades, sequential phosphorylation and signal transduction are all examples of processes where this can apply. The full inclusion of such pathways is seldom advisable when modelling biological systems since the added benefit in dynamical range is outweighed by the disadvantage of an increased model complexity. It is, therefore, common for such pathways to be simplified. However, the manner in which such pathways are simplified is not always particularly effective.

We demonstrate how the coarse-graining of a long linear pathway to a short one (pathway truncation) often lead to a detectably incorrect temporal relationship between the input and the output signal. This discrepancy is important to understand not only because it can cause a model to quantitatively misrepresent time-course data but also because signal timing can qualitatively alter dynamical behaviour. Negative feedback loops, for example, can change from having a stabilising effect to generating oscillations when the feedback is delayed (e.g. [1, 11, 31]). It is, therefore, notable that when this simplification has adverse effects we could identify a detectable signature in the form of homogeneity of the optimised response rate parameters. This can be used as a model diagnostic and could prove especially helpful for complex models where the source of model/data mismatch is not always apparent from their design or output.

Next, we asked whether there might be a way to remedy the shortcoming of a truncated model without increasing the number of model parameters. We propose to assume that the signalling is being passed along the pathway at a constant rate while the number of steps is not fixed. This assumption allows for a three-parameter approximation of arbitrary linear pathways where the pathway length is a tunable parameter. We showed that this 'fixed-rate'

assumption outperformed both truncated pathway models and a DDE model with an explicit time-delay parameter even if the original pathway had highly heterogeneous rates for individual pathway steps. Here, we argue for its effectiveness by applying it to both synthetic and experimental data and we argue that it is applicable for any model input. This was neatly exemplified in Tokuda et al [47] where they showed that introducing distributed delays (fixed-rate assumption) between components of circadian clock models allowed for a parameter reduction while retaining the main dynamics.

A solution of a fixed-rate model allows it to be re-formulated as a gamma-distributed delay. In this model formulation, which does not require a separate differential equation for each pathway step, it is possible to extend the domain of its pathway length parameter, $n$, to real numbers. Doing so improves model accuracy and simplifies some forms of numerical optimisation since this 'gamma model' does not mix integer and real-valued parameters. However, it also disallows the use of ordinary differential equations for numerical simulation which is in many cases more efficient than the alternative. Whether this trade-off is worth it will be dependent on both the case at hand and on the availability of numerical tools for efficient and accurate model simulation.

For a model to be useful, it is important to retain information about the underlying biological system even after the model assumptions have simplified reality. Simplifying assumptions break the precise mapping between model parameters and biological observables. However, much of the connection between the model parameters and real biological processes are retained when the fixed-rate assumption is used for simplification of a linear pathway and even more so when the gamma model is used. If the precise details of a linear pathway are known then the optimal parameters of the gamma model can be obtained from simple expressions. From those expressions, we can, for example, see that optimal pathway length parameter of the gamma model, $n$, provides a lower bound for the length of the real pathway. While the integer-valued pathway length parameter, $n$ of the fixed-rate model may seem more natural when modelling a real pathway with a discrete amount of steps its mapping to reality is actually less precise than that of the gamma model. The mapping is similar to that described for the gamma model, but the relationships described are only approximate for the fixed-rate model. However, this should be contrasted with the truncated pathway model which still uses individual response rate parameters for individual pathway steps. Intuitively, this may seem more closely related to reality but when such a model is optimised to perform the action of a longer pathway, these individual reaction rates become highly decoupled with the response rates of any real pathway step. Rather than being tuned to represent real pathway steps, they are tuned to delay the response peak while not sacrificing too much of the response sharpness. When the model is unable to reproduce the sharpness of the data, its rate parameters become homogeneous. This has the consequence that if two different data-sets are both too sharp for a truncated model, the optimised model will have the same parameters for the two data sets, even if vastly different. Such one-to-many mappings of optimal parameter values to the target data essentially disables inference of biological properties from parameter values. So the fixed-rate assumption not only performs better than the alternatives, but its individual components also retain a closer connection with reality.

A fixed-rate assumption, in line with that proposed to a linear pathway, is applicable also to non-linear systems but its relative merit decreases as the degree of saturation increases. When a kinase cascade is weakly activated (linear) then a fixed-rate assumption provides a more robust and often more effective simplification than the more common approach of disregarding some intermediate steps. When strongly activated, the choice between these simplifications seems to matter less. The non-linearity means that these results are much harder to generalise from since it is no longer true that the model can be completely characterised by its response to a single input. But, even though the performance difference between the two approaches

effectively disappeared under certain conditions, the demonstrated advantage of the fixed-rate assumption under other conditions indicates an over-all benefit from its use.

A mathematical model is defined by its underlying assumptions and can be seen as merely acting as a logical device to deduce the consequences of those assumptions [55]. The main part of model development is thus to find a set of assumptions which accurately and concisely captures the nature of the dynamical system under study. However, finding a good balance between detail and simplicity is often non-trivial and requires some degree of craftsmanship. The better we understand the consequences of specific assumptions, the better we become at striking this balance. In this context, we systematically investigate the consequences of simplifying multi-step pathways by including only a few steps in the model. We clearly demonstrate problems such simplifications may cause. More importantly, we present how to detect these issues when they occur and we provide an alternative approximation to remedy them. We hope this will supply a foundation for well-informed decisions regarding when and how to simplify the ubiquitous linear pathway.

## Methods

### Definition of the linear pathway ODE models

If we model an $n$–step linear pathway that follows Eq 1 except that it receives some input signal to the first step, $X_1$, the dynamical equations can be written as

$$\frac{dX_1}{dt} = \beta_1 \cdot \left( \frac{\alpha_1}{\beta_1} \cdot I(t) - X_1 \right), \tag{18}$$

$$\frac{dX_i}{dt} = \beta_i \cdot \left( \frac{\alpha_i}{\beta_i} \cdot X_{i-1} - X_i \right) \quad \forall i \in \{2, 3, \ldots, n\}. \tag{19}$$

where $\alpha_i$ is a production/activation rate, $\beta_i$ a degradation/deactivation rate, $I(t)$ is some upstream input and $X_n(t)$ is the output of the pathway. The scaling that step $i$ performs on the signal from step $i-1$ is given by $\frac{\alpha_i}{\beta_i}$ and the total scaling, $\gamma$, between the input and output is [28]

$$\gamma = \prod_{i=1}^{n} \frac{\alpha_i}{\beta_i}. \tag{20}$$

Since we are only interested in the dynamics of how the *output* responds to an input we can collapse the scaling into the single parameter $\gamma$ and write

$$\frac{dX_1}{dt} = \beta_1 \cdot (\gamma \cdot I(t) - X_1), \tag{21}$$

$$\frac{dX_i}{dt} = \beta_i \cdot (X_{i-1} - X_i) \quad \forall i \in \{2, 3, \ldots, n\}. \tag{22}$$

Here the position of the $\gamma$ parameter along the pathway is irrelevant for the output. The parameter $\beta_i$ is still a degradation/inactivation rate but the scaling effect that such a parameter usually have has been absorbed into $\gamma$. We, therefore, chose to instead call it a response rate parameter, $r_i \equiv \beta_i$, which better reflects its revised effect on the model. The result is the linear pathway model we have been using in the paper, Eqs 2 and 3.

The analysis of the models was done using different inputs and initial concentrations (Table 1). Unless otherwise stated, the impulse input was used and while it is defined as using the Dirac delta function the actual simulations were done by setting the initial concentration

**Table 1. Model definition of linear pathways with different inputs.** The models are all defined by Eqs 2 and 3, with inputs, $I(t)$, and initial conditions, $\bar{X}(0)$, as specified here.

| Input type | $I(t)$ | $\bar{X}(0)$ |
|---|---|---|
| Impulse | $\delta(t)$ | $(0, 0, \ldots, 0)$ |
| Step | $1$ | $(0, 0, \ldots, 0)$ |
| Ramp | $t$ | $(0, 0, \ldots, 0)$ |
| Wave | $1 + \sin(t)$ | $(\gamma, \gamma, \ldots, \gamma)$ |
| Piecewise | $\begin{cases} 1 & \text{if } 0 < t < 100 \\ 0 & \text{otherwise} \end{cases}$ | $(0, 0, \ldots, 0)$ |
| Noise | Gillespie [56] simulation of: $\begin{cases} \frac{dX}{dt} & = 0.1 + 0.9 \cdot \frac{X}{5+X} - 0.1 \cdot X \\ X(0) & = 0 \end{cases}$ | $(0, 0, \ldots, 0)$ |

of $X_1$ to $\gamma \cdot r_1$. The noisy input was generated using a Gillespie simulation of a single, auto-activating, variable [56]. The noisy time-course that this generated was used as an input both during the generation of synthetic data and during the subsequent model simulations.

## Solutions to the fixed-rate model and the definition of the gamma model

Laplace transforms and the transfer function provides a way of finding a solution to the fixed-rate model with arbitrary inputs [52, 54]. To find this solution, we first assume that $X_i(0) = 0 \; \forall i$ and apply the Laplace transform, $\mathcal{L}$, on both sides on the equations for the fixed-rate model, utilising the fact that the Laplace transform of a function's derivative, $f'$, follows $\mathcal{L}[f'] = s\mathcal{L}[f] - f(0)$. This leads to

$$s\mathcal{L}[X_1] = r \cdot (\gamma \cdot \mathcal{L}[I(t)] - \mathcal{L}[X_1]),$$
$$s\mathcal{L}[X_i] = r \cdot (\mathcal{L}[X_{i-1}] - \mathcal{L}[X_i]),$$

which can be rearranged to get

$$\mathcal{L}[X_1] = \frac{\gamma \cdot r}{s + r} \cdot \mathcal{L}[I(t)],$$
$$\mathcal{L}[X_i] = \frac{r}{s + r} \mathcal{L}[X_{i-1}].$$

This can be recursed to solve for $n$ steps, leading to

$$\mathcal{L}[X_n] = \frac{\gamma \cdot r^n}{(s + r)^n} \cdot \mathcal{L}[I(t)].$$

From this, we can easily get the transfer function, $G(s)$, of the fixed-rate model,

$$G(s) = \frac{\mathcal{L}[X_n(t)]}{\mathcal{L}[I(t)]} = \frac{\gamma \cdot r^n}{(r + s)^n}. \tag{23}$$

The impulse response function, $g(t)$, is the inverse Laplace transform of this transfer function, given by

$$g(t) = \mathcal{L}^{-1}[G(s)] = \frac{\gamma \cdot r^n \cdot t^{n-1} \cdot e^{-rt}}{(n - 1)!}. \tag{24}$$

This impulse response can be used to get a solution to $X_n(t)$ for almost any input (the input needs to *have* a Laplace transform, even if we never have to calculate it). From Eq 23, we have that

$$\mathcal{L}[X_n(t)] = G(s)\mathcal{L}[I(t)].$$

This equation is still in the complex domain but we can use the connection between multiplication in the complex domain to convolution in the time domain to get the desired function

$$\mathcal{L}[X_n(t)] = G(s)\mathcal{L}[I(t)],$$
$$\Leftrightarrow,$$
$$X_n(t) = \int_0^t I(t')g(t-t')dt', \tag{25}$$
$$= \int_0^t I(t') \frac{\gamma \cdot r^n \cdot (t-t')^{n-1} \cdot e^{-r \cdot (t-t')}}{(n-1)!} dt'.$$

In order to improve the model's ability to fit data, we can replace $(n-1)!$ with $\Gamma(n)$. This makes no difference for integer $n$ but it expands the possible domain of $n$ to all real numbers greater than or equal to 1. After this, we end up with what we call the gamma model which can be expressed in either input-specific forms (e.g. Eq 11) or in its gamma-distributed delay form (Eq 12).

## Model simulation

Differential equations were solved using an algorithm with stiffness detection that toggled between Tsitouras5 for non-stiff regions and Rosenbrock23 for stiff regions [57–59]. The integral of the gamma-distributed delay model (Eq 12) was evaluated using Gauss-Kronrod quadrature.

For the wave input we needed an initial concentration of $X_i(0) = \gamma \; \forall i$. However, since we in the definition of the model assumed an all-zero initial concentration of the pathway, it is built-in to the model that it starts from an all-zero state. This limitation can, however, often be circumvented by performing the simulation in two stages. The first stage would apply the input required for the model to get to the proper 'initial' conditions for the second stage which simulates the model with the input one actually wished to study. For the wave input in this paper, this was achieved by using the input function

$$I(t) = \begin{cases} 1, & t < 0 \\ 1 + \sin(t), & t \geq 0 \end{cases}$$

and by allowing the integration to start at a negative $t$. The actual value used was based on the signal delay and the duration of the gamma model

$$t_{min} = t - \tau - 3 \cdot \sigma = -\frac{n}{r} - 3 \cdot \frac{\sqrt{n}}{r}.$$

This discards the effect of inputs that were early enough to have a very small impact on the current output value.

## Data generation

Synthetic data sets were generated using the fixed-step model (Eqs 2 and 3). The procedure was to first draw an integer between 1 and 50 to be used as the number of pathway steps, $n_{data}$. A response rate, $r_i$ was randomly drawn for each of the $n_{data}$ steps in the pathway. For most

input types, the parameters $r_i$, were generated by transforming the uniformly random variable $Y_i \sim U(-2, 1)$ according to $r_i = 10^{Y_i}$. However, since any too slow reaction rates will filter out the dynamics of the noise and the wave input, the reaction rates for those models were drawn from $r_i \sim 10^{\mathcal{U}(-1,1)}$ and $r_i \sim 10^{\mathcal{U}(0,1)}$, respectively. The value of the scaling parameter, $\gamma$, was in all cases set to 1. The model was then run and the resulting trajectory of the last pathway step, $X_n(t)$, was stored for use as the synthetic data set (5000 times for each input).

A similar scheme was used for the non-linear cascade model. There, 2000 sets of parameters were drawn according to $n_{data} \sim \mathcal{U}\{1, 20\}$, $\alpha_i \sim 10^{\mathcal{U}(-0.5,1.5)}$, $\beta_i \sim 10^{\mathcal{U}(-1,1)}$, and $\gamma_i \sim 10^{\mathcal{U}(-1,1)}$.

### Fitness definition

In order to automatically evaluate the fitness of a model, an 'integral cost' function was defined. The idea of this cost function is to measure the mismatch in the area under the curve for the model and the data (Fig 12). This is a form of $\ell1$ norm that ensures an even sampling of the data for cost evaluation. We define this cost value as

$$C = \frac{\int_{-\infty}^{\infty} |X_{model}(t) - X_{data}(t)| dt}{\int_{-\infty}^{\infty} |X_{data}(t)| dt}. \tag{26}$$

Numerically, $C$ was calculated using Riemann sums and interpolations of both the data and the model solution. The evaluations started at $t = 0$ and for the 'Impulse', 'Step', and 'Piecewise' input types they continued until the model derivatives were close to zero (absolute and relatives tolerance $10^{-8}$ and $10^{-6}$, respectively) or until $t = 5000$, whichever came first. The 'Piecewise' input simulations were additionally prohibited from stopping before the end of the piecewise constant input. Since the 'Ramp', 'Wave' and 'Noise' inputs do not allow for equilibration, we set fixed stopping times of $t = 500$, 30 and 200, respectively.

Much of the analysis was also repeated using the more commonly used normalised least square cost function. While the results are slightly different, they did not change any of the conclusions in this work.

### Model optimisation

The model parameters were all optimised to reproduce the time-trajectory of each synthetic data set. The optimisation target was to minimise the cost value, $C$, described above. For the

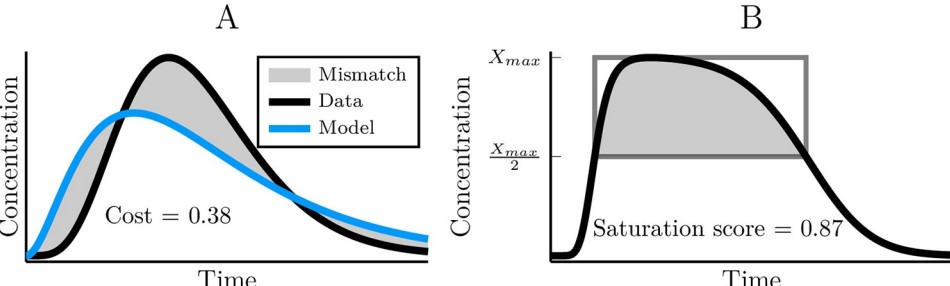

**Fig 12. Demonstrating the cost value and saturation score.** (A) The cost value measures the area mismatch between the model output and that of the data. Linear models used the area under the data curve for normalisation and the kinase cascade models used the union of the data and model areas. (B) The saturation score indicates the degree to which a kinase cascade is saturated. If the curve is the output of the system, the saturation score is the fraction of the indicated square that is filled by the shaded region under the curve.

**Table 2. Parameter search space for the optimisation of the different models.**

| model | $\gamma$ | $r$ | $n$ | $\tau$ |
|---|---|---|---|---|
| fixed-step | $[10^{-3}, 10^3]$ | $[10^{-3}, 10^3]$ | N/A | N/A |
| fixed-rate | $[10^{-1}, 10^1]$ | $[10^{-2}, 10^2]$ | $\{1..30\}$ | N/A |
| gamma | $[10^{-4}, 10^2]$ | $[10^{-3}, 10^3]$ | $[1, 31]$ | N/A |
| DDE | $[10^{-4}, 10^2]$ | $[10^{-4}, 10^1]$ | N/A | $[10^{-2}, 10^4]$ |

actual optimisation, we used an adaptive differential evolution algorithm following the /rand/ 1/bin/ scheme, with a radius limited sampler that took 2,000 steps per free parameter of the model [60, 61]. The search space for the parameters of the different models were chosen to allow for the time delays that the synthetic data could generate (Table 2). The sampling space was linear for *n* and logarithmic for the other parameters.

## Non-linear analysis

The non-linear kinase cascade models were defined by Eqs 16 and 17, as described in Results. A pulse input was applied to initially inactive cascades for both the data generation and the model simulations,

$$I(t) = \begin{cases} \xi & \text{if } 0 < t < 1 \\ 0 & \text{otherwise.} \end{cases} \tag{27}$$

Here, $\xi$ was set to 1 except for when it was used to scale the input, as stated in Results. 2000 data sets were generated where the data consisted of a single trajectory for a single input and 500 were generated where each data set consisted of three trajectories, for three differently scaled inputs. Simulations were done using DifferentialEquations.jl's Rosenbrock23 ODE solver [58, 59]. A few of the proposed parameter sets for data-generation caused numerical errors in the differential equation solver and were re-drawn.

The normalisation of the cost function was revised for the non-linear study to ensure that the cost was symmetric in terms of whether the data or the model curve was the largest one. The revised cost function was defined by

$$C = \frac{\int\limits_{-\infty}^{\infty} |X_{model}(t) - X_{data}(t)| dt}{\int\limits_{-\infty}^{\infty} \max\left(X_{data}(t), X_{target}(t)\right) dt}. \tag{28}$$

The number of optimisation steps per parameter was increased in the non-linear case to 5, 000. The parameter search range is defined in Table 3. The fixed-rate model had a smaller search range since the repeated application of steps with very high signal gain or dampening were neither relevant nor numerically stable. Some parameter sets proposed by the optimiser resulted in prematurely terminated simulations due to numerical issues. When this happened during optimisation, the parameter set was rejected by assigning it an infinite cost. Such errors

**Table 3. Parameter search space for the optimisation the non-linear kinase cascade models.**

| Model | $\alpha$ | $\beta$ | $X^{tot}$ | $n$ |
|---|---|---|---|---|
| Fixed-step | $[10^{-8}, 10^5]$ | $[10^{-8}, 10^5]$ | $[10^{-8}, 10^5]$ | N/A |
| Fixed-rate | $[10^{-2}, 10^2]$ | $[10^{-3}, 10^3]$ | $[10^{-6}, 10^1]$ | $\{1..20\}$ |

also occurred during the analysis of model predictiveness (Fig 11) and when it did the offending parameter set was excluded from the analysis (8% for the two-step model and 3% for the fixed-rate model).

We defined a 'saturation score' to reflect the degree to which the dynamics of a cascade was dominated by saturation. For this, we examined the top half of the output signal peak and divided its area with that of a square which encompasses this half peak (Fig 12B).

## Software availability

All the computational results were generated using the Julia programming language [62]. Differential equations were solved using DifferentialEquations.jl, Gauss Kronrod quadrature was done with QuadGK.jl and optimisation was done with BlackBoxOptim.jl [59, 61]. Source code for the reproduction of the results is openly available under the MIT licence at the Sainsbury Laboratory GitLab repository https://gitlab.com/slcu/teamHJ/publications/Korsbo_et_al_2020. This repository also includes documentation and a tutorial aimed at making reproduction and reuse easy.

## Supporting information

**S1 Fig. Fitting one-step models to data from linear pathways of different length.** (A-E) The worst model/data fits for a given length, $n_{data}$, of the model that generated the data. Coloured lines show simulations of the fitted model while black lines show the synthetic data. (F) The cost value for 5000 parameter sets, each optimised towards a different set of synthetic data. Circles show the cost values resulting from data wherein all the steps in the data-generating linear pathway were randomly drawn, $r_i \sim 10^{U(-2,\ 1)} \forall i$. Stars are the cost values from data wherein the pathway has homogeneous reaction rates, $r_i = 1 \forall i$. The x-axis shows the number of steps in the model which were used to generate the data. (G-K) Examples of the best model/data fits for different data pathway lengths, $n_{data}$.
(TIF)

**S2 Fig. Fitting three-step models to data from linear pathways of different length.** Description otherwise as in S1 Fig.
(TIF)

**S3 Fig. Fitting four-step models to data from linear pathways of different length.** Description otherwise as in S1 Fig.
(TIF)

**S4 Fig. Fitting five-step models to data from linear pathways of different length.** Description otherwise as in S1 Fig.
(TIF)

**S5 Fig. Fitting gamma models to data from linear pathways of different length.** The gamma model (Eq 11) with $n_{model} \in \mathbb{R}$ efficiently recapitulates linear pathway dynamics. (A-E) Examples of the worst model-data fits for different lengths of the data-generating pathway, $n_{data}$. Black lines show the synthetic data, orange lines show the results of the gamma model, and grey lines show the results of a two-step model (Eqs 2 and 3). (F) The ability of the gamma model to fit the data changes with the length of the underlying pathway, $n_{data}$. Blue dots show the cost value of the gamma model. The small, grey, dots show the corresponding cost values for a two-step model. (G-K) Examples of the best model-data fits.
(TIF)

**S6 Fig. A comparison of the cost values when using either the fixed-rate model (Eqs 8 and 9) or a fixed-step model (Eqs 2 and 3) to fit the same data.** Each circle represents a single synthetic data set. 20 synthetic data sets were generated for each $n_{data} \in \{1, \ldots, 50\}$ using the impulse input (Table 1). The circles position indicates the optimised cost for the respective models and its color indicates the $n_{data}$ value of the data-generating pathway. Percentages indicate how many of the data sets had a higher (worse) cost value for the respective models. (TIF)

## Acknowledgments

We would like to thank Torkel Loman and members of the Jönsson group for fruitful discussions; Torkel Loman, Ross Carter and James Locke for feedback on the manuscript; and Alexey Chizh who's masters project on the simplification of different pathways never made it into the paper but was still informative.

## Author Contributions

**Conceptualization:** Niklas Korsbo, Henrik Jönsson.

**Data curation:** Niklas Korsbo.

**Formal analysis:** Niklas Korsbo.

**Investigation:** Niklas Korsbo.

**Methodology:** Niklas Korsbo, Henrik Jönsson.

**Project administration:** Henrik Jönsson.

**Resources:** Henrik Jönsson.

**Software:** Niklas Korsbo.

**Writing – original draft:** Niklas Korsbo.

**Writing – review & editing:** Niklas Korsbo, Henrik Jönsson.

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
