## [Decision Letter · Decision Letter 0]

7 Jan 2020

Dear Dr Jonsson,

Thank you very much for submitting your manuscript 'It's about time: Analysing an alternative approach for reductionist modelling of linear pathways in systems biology' for review by PLOS Computational Biology. Your manuscript has been fully evaluated by the PLOS Computational Biology editorial team and in this case also by independent peer reviewers. The reviewers appreciated the attention to an important problem, but raised some substantial concerns about the manuscript as it currently stands. While your manuscript cannot be accepted in its present form, we are willing to consider a revised version in which the issues raised by the reviewers have been adequately addressed. We cannot, of course, promise publication at that time.

Sincerely,

Christopher V. Rao

Associate Editor

PLOS Computational Biology

Douglas Lauffenburger

Deputy Editor

PLOS Computational Biology

[LINK]

Reviewer's Responses to Questions

**Comments to the Authors:**

Reviewer #1: General comment:

In biochemical network modeling, it is a common practice to replace a long sequence of reactions with fewer reaction steps. Such simplification often involves lumping and rescaling of rate parameters to keep the reduced network consistent with the original system. This manuscript points out the caveats of such simplification and provides a few alternatives. The main analyses and findings of this work are summarized below.

The manuscript considers a linear pathway consisting of N reaction steps. It also considers another linear pathway consisting of n steps (n \\neq N) that serves as a model for the N-step pathway. By fitting the n-step model to (synthetic) data produced by the N-step pathway, the authors investigate how well the model recapitulates the system’s dynamic behaviors in response to several inputs. It appears that the model performs poorly when n << N (most analyses involve N = 50, n = 2), especially when the steps are assigned with similar or identical rates. The authors then propose an alternative model, where n is treated as a free variable whose value is determined by optimization while the reaction rates of all steps are held fixed. In the majority of the cases, this alternative model (fixed-rate model) outperforms a two-step model. Finally, the authors provide an analytical form for the fixed-rate model for a simple case where the system input is zero. This analytical form, called the beta model, is not subject to the constraint that the number of steps n must be an integer. This beta model outperforms both the fixed-rate and the two-step model.

The analyses are interesting and the reviewer thinks the manuscript is well-written. Nevertheless, I have several questions, as listed below.

Major comments:

1) The equations indicate that the authors only consider first-order reactions in describing their linear pathway. However, most biochemical transformations (steps) are better described by the Michalis-Menten rate law. For example, a phosphorylation reaction involves a transient complex formation between the substrate protein and an enzyme (kinase). Such steps are prevalent in the examples provided by the authors in the introduction: MAPK activation, transcription or translation by RNA polymerase or ribosome, sequential protein phosphorylation in various pathways, etc. Could the authors provide examples of published models that treated these steps as first-order reactions?

Are the analyses and conclusions still valid if the Michaelis-Menten rate law is used to describe the steps in the sequential pathway?

2) The analyses provided are based on generic or toy examples. Better if the authors provided a case study involving a specific biological pathway. Examples could be published models where such simplification was applied.

Most analysis involve a 50-step vs. a 2-step model (N = 50 vs. n = 2). A 50-step linear pathway without having an intermediate crosstalk/non-linear component/feedback seems unlikely in cell signaling or gene transcription. The result indicates the two-step model performs quite well when N < 10, and the two-step model is almost as good as the fixed-rate model in the range N =5 -10 (Fig. 4). In a more realistic scenario, where N < 10, is the two-step model adequate?

3) The optimization evaluated the number of steps for the fixed-rate model. This should lead to many more steps in the fixed-rate model compared to the two-step model when N = 50 (Fig. 5 and 6). Is not it expected that the fixed-rate model outperforms the two-step model because the former involves more steps? The analysis shows that the fixed-rate model outperformed the 2-step model in approximately 90% of the cases. How would this result change if the fixed-rate model was compared against a 5-step model? Or if the data was generated using a 10-step pathway (N = 10)?

4) “---- it is easy to find examples where it has been used to simplify multi-step reactions such as protein production (e.g. 7, 31{33); protein-to-protein signalling networks (e.g. 7, 32{34); protein modifications such as 57 phosphorylations, methylations, and ubiquitinations (e.g. 33, 34)”

--- It seems like these simple models (cited references) did not truncate or simplify a linear multistep reaction, rather the goal could be to use a minimal model to describe specific data or experimentally-observed phenomena. Are these citations relevant in the context of this work?

Minor comments:

1) In the DDE model, the delay time is enforced in the final step. Does it make any difference if the optimization is allowed to choose where to introduce the delay time? What if more steps are allowed with fixed or heterogeneous delay times?

2) Font sizes in the legends of Fig. 5d and 6E are too small to read.

3) In the result section, figures are often referenced randomly without following any specific order.

Reviewer #2: This paper focuses on the impact of number of reaction steps in linear biochemical pathways. They introduce a framework for linear pathway simulations in order to compare several reduction operations over such pathways : reduction of the number of reaction steps, introduction of several classes of delays. Based on their simulations on synthetic dataset, the author conclude that simplifying to a three-model parameters (more precisely, scaling, pathway lenght, homogeneous response rate of the pathway) outperforms both state-of-the-art reduction approaches.

The paper is well-written, interesting and technically correct. However, I have doubts on its applicability on real models.

The first issue raised by the paper is that the main assumption of the author is that is a linear chain of reactions can be modeled with a linear differential system, without taking into account any non linearity effects. The study performed by the authors concerns only the biological systems for which this assumption is true. The paper would be more convincing if the author could detail the number of biological models (for instance in the BioModels database) for which this linear model assumption is used.

A second issue is that the synthetic data used to conduct all the experimentation were generated according to a linear ODE model with different pathway lenghts (from 1 to 50) and a log-uniform distribution for response rates. How valid are these hypotheses with respect to existing ODE models of biological processes processes in the litterature ? Simulating several linear published models to provde that the synthetic data are realistic seems necessary to validate the approach.

A third issue is that, as mentionned by the authors, linear pathways occur in biological models as combined with self-regulatory controls that may impact on the input signal. Therefore, the model of the input signal should deserve a very special attention to validate the conclusion of the authors. Addressing this issue could be done for instance by applying the three reduction methods studied in the paper on a family of complex published and validated models in order to figure out if the conclusions are still valid for input functions controlled by the system dynamics.

Finally, the author advocate that a software is associated with their publication. It appears that this software is a code in the Julia programming language allowing to reproduce the simulations shown in the paper. It should be clarified that the software is not a tool for model reduction (and parameter fitting) as it could be expected while reading the paper.

Reviewer #3: The manuscript addresses the problem of the number of steps in a simplified, linear model of signaling with no branching. This is an important issue and may guide the modeler’s choice. The main result of the paper is that a “fixed rate” model with a variable number of steps is well suited for fitting a broad corpus of data. This idea has already been used in the systems biology literature but never tested systematically. The submitted work has the merit of testing this reduction ansatz. Nevertheless, the conclusions on the applicability of the “fixed rate” ansatz are entirely based on numerical simulations instead of precise mathematical estimates. In many places, the manuscript can be substantially improved by precise definitions and more rigorous specification of the domain of validity of the results. A list of issues that need major amendments follows:

1) The idea to use the number of steps as a free parameter has already been used for a chain model of transcription in Dufourt et al Nature comm. (2018) 9: 5194, where the authors chose the number of steps and the homogeneous or heterogeneous rates on the bases of quality of fitting and variability of optimal and suboptimal parameters; for the optimal number of steps a heterogeneous rate hypothesis is rejected on the basis of parameter variability. These similar ideas should be cited in the manuscript.

2) The type of model and the meaning of linearity should be made more precise in summary and introduction. Which type of models are dealt with, stochastic or deterministic? The authors use the name “linear pathways” to speak of a special case of first order chemical reaction networks (CRNs), first introduced by Heinrich et al (2002) ref. [12]. In this context linearity may mean both “first-order” and unbranched topology.

3) The notion of “step” is crucial in the paper and needs careful discussion.

4) It is not true that little effort has been made to understand pathway truncation. Mathematical theories of pathway truncation are available for monomolecular CRNs (Gorban and Radulescu Adv.Chem.Eng. (2008) 34:103), first order CRNs (Helfferich J.Phys.Chem. (1989) 93:6676), and more general CRNs (Radulescu et al, Front. Genet. (2012) 3:131).

5) The scaling leading from Eqs (1) to (2) and (3) should be explicitly given in the main text (this is not available in the methods).

6) Clearly specify that the output is Xn(t).

7) Eq. at line 120 is valid only if Xi(0)=0 for i >1.

8) At various places fitting is said to be guaranteed perfect. But fitting results from a numerical scheme; even if theoretically a perfect fit is possible, the scheme may not find it. At other places fitting is said to perform well. What does this mean quantitatively and on which statistical grounds?

9) Some criteria are proposed to identify “detrimentally truncated models”, such as the amplitude and width if the output signal and the variability in the rate parameters. This section is important and should be treated with more care. What quantitative recipes should be used here?

10) The eq (6) corresponds to a singular delay model, whereas the fixed rate model leads to a distributed delay model. This should be clearly stated. The explanation of the failure of the singular delay model (lines 218-220) is obscure.

11) The r parameter of the fixed rate model is said to represent the slowest step. This is rigorously true for separated rates but not necessarily true for similar rates. I don’t think that Fig4 represents a rigorous proof of this correlation. Computing the correlation with next slowest steps and with the harmonic mean and comparing them with the slowest step would be a proof.

12) A gamma model with real shape parameter is proposed. This implies non-integer number of steps. What does it mean?

13) The way how the restriction on the zero initial concentration can be lifted (lines 297-300) is not clearly explained.

14) In the conclusion (also in the introduction) it is said that wrong delays can destabilize oscillations. To which extent is this true? Can the authors provide examples where delays with the same mean, but different distributions have different effect on the stability of oscillations? Some results from delay differential equations could be invoked.

15) Methods should be proofread. Eq.11 seems to be used for proving a number of statements that are by no means clear. What is the meaning of the linear dependence of the production terms (line 380)? The whole paragraph between lines 378 and 390 is obscure. Same for lines 438-443, 447-450 (the l1 norm does not have the same advantages?). What are the “interpolations” at line 452?

16) How is the heterogeneity parameter delta defined for the general distribution of rates? The definition in the figures is based on equidistant rates in log scale, which is a very special choice.

**Have all data underlying the figures and results presented in the manuscript been provided?**

Reviewer #1: None

Reviewer #2: Yes

Reviewer #3: No: The conclusions of the manuscript rely heavily on numerically generated data. This data should be made available on a public repository such as zenodo, for instance.

PLOS authors have the option to publish the peer review history of their article (what does this mean?). If published, this will include your full peer review and any attached files.

Reviewer #1: No

Reviewer #2: No

Reviewer #3: No

---

## [Decision Letter · Decision Letter 1]

27 May 2020

Dear Dr Jonsson,

We are pleased to inform you that your manuscript 'It's about time: Analysing simplifying assumptions for modelling

multi-step pathways in systems biology' has been provisionally accepted for publication in PLOS Computational Biology. The reviewers were mixed regarding its suitability for PLOS Computational Biology. Based on our reading of the manuscript, however, we believe that it is indeed suitable. 

Best regards,

Christopher V. Rao

Associate Editor

PLOS Computational Biology

Douglas Lauffenburger

Deputy Editor

PLOS Computational Biology

Reviewer's Responses to Questions

**Comments to the Authors:**

Reviewer #1: I am satisfied with the revision made by the authors in this current manuscript. One of my primary concerns was that the scope of this work could be limited to a linear system only. In this revised manuscript, the authors have demonstrated applications to non-linear systems involving the Michaelis-Menten reaction scheme. This change should broaden the scope of the work as signaling pathways mostly represent nonlinear systems. I also found other revisions in the manuscript satisfactory based on my remaining suggestions.

Reviewer #2: The authors have substantially changed their manuscript to include additional theoretical proofs about systems equivalence.

The main issue raised in my first review was that the application of the proved results may have a limited impact if the authors could not be able to provide a panel of examples of models on which the reduction they suggest could apply. Unfortunately, the authors added a reference to a recent paper (2019) and a validation on a toy-example model of Aradpidopsis Thailand but did not really apply their approach to a convincing panel of existing model. On the contrary, they argue that general theoretical proofs are more convincing than applications to real middle-scale examples.

As a consequence, the author have substantially changed the manuscript to include theoretical proofs to justify the reduction methods. It seems to me that the present version of the manuscript does not fit for PLOS computational Biology and would deserve a more theoretical and mathematical audience.

**Have all data underlying the figures and results presented in the manuscript been provided?**

Reviewer #1: None

Reviewer #2: None

PLOS authors have the option to publish the peer review history of their article (what does this mean?). If published, this will include your full peer review and any attached files.

Reviewer #1: No

Reviewer #2: No

---

## [Editor Report · Acceptance letter]

23 Jun 2020

PCOMPBIOL-D-19-01662R1 

It's about time: Analysing simplifying assumptions for modelling
multi-step pathways in systems biology

Dear Dr Jönsson,

I am pleased to inform you that your manuscript has been formally accepted for publication in PLOS Computational Biology. Your manuscript is now with our production department and you will be notified of the publication date in due course.

With kind regards,

Sarah Hammond
